# Identification of an oncogenic network with prognostic and therapeutic value in prostate cancer

Fiorella Magani[1,2,†], Eric R Bray[3,†], Maria J Martinez[2], Ning Zhao[2], Valeria A Copello[1,2] (iD), Laine Heidman[2], Stephanie O Peacock[1,2], David J Wiley[2], Gennaro D'Urso[2] & Kerry L Burnstein[2,4,*] (iD)

## Abstract

Identifying critical pathways governing disease progression is essential for accurate prognosis and effective therapy. We developed a broadly applicable and novel systems-level gene discovery strategy. This approach focused on constitutively active androgen receptor (AR) splice variant-driven pathways as representative of an intractable mechanism of prostate cancer (PC) therapeutic resistance. We performed a meta-analysis of human prostate samples using weighted gene co-expression network analysis combined with experimental AR variant transcriptome analyses. An AR variant-driven gene module that is upregulated during human PC progression was identified. We filtered this module by identifying genes that functionally interacted with AR variants using a high-throughput synthetic genetic array screen in *Schizosaccharomyces pombe*. This strategy identified seven AR variant-regulated genes that also enhance AR activity and drive cancer progression. Expression of the seven genes predicted poor disease-free survival in large independent PC patient cohorts. Pharmacologic inhibition of interacting members of the gene set potently and synergistically decreased PC cell proliferation. This unbiased and novel gene discovery strategy identified a clinically relevant, oncogenic, interacting gene hub with strong prognostic and therapeutic potential in PC.

**Keywords** androgen receptor splice variants; castration resistance; mitotic gene signature; weighted gene co-expression network analysis; yeast synthetic genetic array
**Subject Categories** Cancer; Genome-Scale & Integrative Biology; Network Biology
**Mol Syst Biol. (2018) 14: e8202**

## Introduction

Combining computational systems-level analyses of patient samples with rigorous experimental high-throughput approaches is a powerful, unbiased strategy to identify novel biomarkers and therapeutic targets in cancer. Such approaches generate gene co-expression modules that permit researchers to uncover disease-relevant, dynamic gene networks (e.g., Lamb *et al*, 2006; Zhu *et al*, 2016). Identified gene clusters (co-expression modules) often represent genes involved in common biological or pathologic functions, genes that define cell-type differences, and/or genes governed by a common regulatory mechanism (e.g., driven by a specific transcription factor). Identifying the components of gene modules that control key pathways of cancer progression and therapeutic resistance permits pathway targeting, as opposed to single oncogene targeting, which readily leads to therapeutic resistance (Kreeger & Lauffenburger, 2009). We applied such an approach to PC, which is the second-leading cause of cancer-related death in U.S. men (Siegel *et al*, 2016).

Androgen deprivation therapy (ADT) is the standard of care for non-organ confined advanced PC, leading to tumor regression. However, PC inevitably recurs as castration-resistant prostate cancer (CRPC; e.g., Knudsen & Penning, 2010; Karantanos *et al*, 2013), an incurable form of the disease with limited therapeutic options (Antonarakis *et al*, 2014). A proposed major driver of CRPC progression is constitutively active AR variants (AR-Vs) that lack the C-terminal ligand-binding domain (LBD) of the full-length AR, but retain the potent transactivating N-terminal domain (NTD) and the DNA-binding domain (DBD). These AR-Vs are thought to constitutively promote transcription of an oncogenic program resulting in therapeutic resistance (Zhang *et al*, 2011; Li *et al*, 2013; Antonarakis *et al*, 2017; Ho & Dehm, 2017). AR-V7 (also termed AR3 or AR1/2/3/CE3) is the best-characterized AR splice variant in human specimens (Hörnberg *et al*, 2011). AR-V7 homodimerizes and heterodimerizes with full-length AR resulting in constitutive, unregulated AR activity (Watson *et al*, 2010; Chan *et al*, 2015; Xu *et al*, 2015). Additionally, the expression of AR-V7 is linked to poor prognosis, epithelial–mesenchymal transition (EMT; Kong *et al*, 2015),

1   Sheila and David Fuente Graduate Program in Cancer Biology, University of Miami Miller School of Medicine, Miami, FL, USA
2   Department of Molecular and Cellular Pharmacology, University of Miami Miller School of Medicine, Miami, FL, USA
3   Department of Neurological Surgery, Miami Project to Cure Paralysis, University of Miami Miller School of Medicine, Miami, FL, USA
4   Sylvester Comprehensive Cancer Center (SCCC), Miami, FL, USA
    *Corresponding author. Tel: +1 305 243 3299; Fax: +1 305 243 4555; E-mail: KBurnstein@med.miami.edu
    †These authors contributed equally to this work

and resistance to current treatments (Qu *et al*, 2015; Antonarakis *et al*, 2017). Even though AR-V7 is considered a plausible target for CRPC therapy, the design of specific high-affinity compounds is a major challenge due in large part to the lack of the LBD (reviewed in Imamura & Sadar, 2016).

Gene networks downstream of AR-V7 drive its oncogenic signaling and disease progression (e.g., Hu *et al*, 2012; Cao *et al*, 2014; Shafi *et al*, 2015; Xu *et al*, 2015). Identifying disease-relevant AR-V7-driven genes is challenging, in part because of the extensive overlap with the full-length AR transcriptome (e.g., Watson *et al*, 2010; Chan *et al*, 2012; Hu *et al*, 2012; reviewed in Lu *et al*, 2015). Components of this network may not only underlie AR-V7 oncogenicity but possess the dual feature of enhancing AR activity, since positive feedback loops are common in endocrine cancers (including PC). Such reciprocally acting proteins have the strongest potential to be exploited therapeutically (e.g., Chia *et al*, 2011; Karacosta *et al*, 2012; Goodwin *et al*, 2015). Moreover, there is a need for prognostic biomarkers of tumor aggressiveness that can stratify patients at the time of PC diagnosis and thereby guide clinical management accordingly by predicting the patient's therapeutic response to treatment (Bjartell *et al*, 2011; Prensner *et al*, 2012).

We utilized a large cohort of publicly available gene microarray data to identify gene clusters associated with PC and CRPC and then combined this information with rigorous experimental approaches in different model systems to identify disease-relevant genes. With this gene discovery strategy, which is broadly suitable, for example, for any disease driven by a transcription factor, we defined a novel gene set that served as a prognostic marker for PC and revealed a new drug combination that potently and synergistically inhibited CRPC growth.

## Results

We performed weighted gene co-expression network analysis (WGCNA) to identify, in an unbiased manner, gene modules associated with different types/stages of PC pathologies and phenotypes. WGCNA is based on the concept that co-expressed genes across a series of traits (in this instance, pathological features of human prostate) share biological functions and/or are controlled by a common mechanism, such as by a specific transcription factor(s) (Kadarmideen & Watson-Haigh, 2012). Thus, genes co-expressed across patient samples are clustered together in a module. For this meta-analysis, we used eight publically available microarray datasets that utilized the same array platform (Appendix Fig S1A) and encompassed six different prostate phenotypes/disease stages (Appendix Fig S1B). The microarray datasets were combined and used for network construction (Appendix Fig S1C). Gene modules were first assembled with a minimum module size of 30 genes, and highly similar modules were combined using a dissimilarity threshold of 0.25 resulting in 20 gene modules that were then correlated to different prostate phenotypes (Fig 1A). Three of the 20 modules (arbitrarily termed: *green, magenta*, and *yellow)* contained genes whose expression levels had significant positive associations with PC and CRPC (Fig 1A and Appendix Fig S1D).

To determine whether any of the WGCNA modules were enriched for genes regulated by AR-V7, we performed gene expression profiling in the human CRPC cell line 22Rv1. We chose 22Rv1

cells since, even though they express full-length AR, they contain high levels of AR-V7 and depend on AR-V7 for growth and survival (Dehm *et al*, 2008; Guo *et al*, 2009; Marcias *et al*, 2010). We performed doxycycline-inducible knockdown of AR-V7 using a specific tet-pLKO shAR-V7 construct (Appendix Fig S2A and Peacock *et al*, 2012). We then mapped the resulting 3,645 AR-V7-regulated genes to the WGCNA modules. Strikingly, nearly 75% of the *green* module genes (equivalent to 60 genes) exhibited decreased expression following AR-V7 knockdown (i.e., were upregulated by AR-V7; Fig 1B and Appendix Fig S2B). In contrast, there was no significant enrichment of genes regulated under the control condition (shGFP) with any of the WGCNA modules (Appendix Fig S2C). The *green* module was highly enriched in genes associated with cell proliferation, particularly mitotic cell cycle and chromosome segregation (Appendix Fig S2D). This module contained a number of genes previously linked to prostate or other types of cancers including *RAD51, AURKA, CENPE, EZH2, TOP2A, BUB1, TPX2, CDK1,* and *CCNB1*. Because of the overlapping transcriptomes of full-length AR and AR-V7, we examined whether full-length AR also regulated genes in the *green* module. We utilized two full-length AR-regulated gene signatures: One consisted of genes differentially expressed in tumor versus normal samples and enriched for AR binding sites, obtained from Pomerantz *et al* (2015), and a second transcription-based full-length AR activity signature from Mendiratta *et al* (2009). We then examined the distribution of these full-length AR-regulated genes across the WGCNA modules (Appendix Fig S3A and B). We found that no gene in the *green* module was regulated by full-length AR using the full-length AR signature from Mendiratta *et al*, and only 6% of genes in the *green* module were full-length AR targets based on the Pomerantz *et al* dataset. These results suggest that the *green* module is largely and selectively regulated by AR-V7, but not full-length AR. Thus, we identified 60 genes (nearly 75% of the *green* module) regulated by AR-V7, whose expression was associated with and upregulated in PC, CRPC, and metastasis in the WGCNA meta-analysis of human samples.

This set of 60 clinically relevant genes, which are regulated by AR-V7, can be further analyzed in a number of ways to understand the mechanisms of AR variant action in PC. In this particular study, we were interested in identifying those genes that also interact in a biologically relevant way and might participate in a positive feedback loop to enhance AR signaling. Such genes are likely to encode key prognostic markers as well as potential therapeutic targets acting within the AR-V7 network.

To identify such genes, we generated an AR-V7 functional genetic *interactome* using a high-throughput synthetic genetic array (SGA) screening method in the yeast *Schizosaccharomyces pombe*. This unbiased and powerful approach has successfully identified other human disease-related protein interactomes [e.g., for X-linked spinal muscular atrophy (SMA) (Wiley *et al*, 2014)]. Using the methods described in detail in Wiley *et al* (2014), we generated an inducible *S. pombe* strain expressing an HA-tagged AR-V7 fusion protein integrated under the control of the *nmt1* thiamine-repressible promoter. While expression of AR-V7 generated a slight growth defect, the strain achieved growth saturation and readily allowed for the SGA screening (Appendix Fig S4). The AR-V7 strain was then crossed with an *S. pombe* gene deletion library to create 3,664 unique gene deletion strains that inducibly express AR-V7. Functional genetic interactions ("hits") were inferred when the

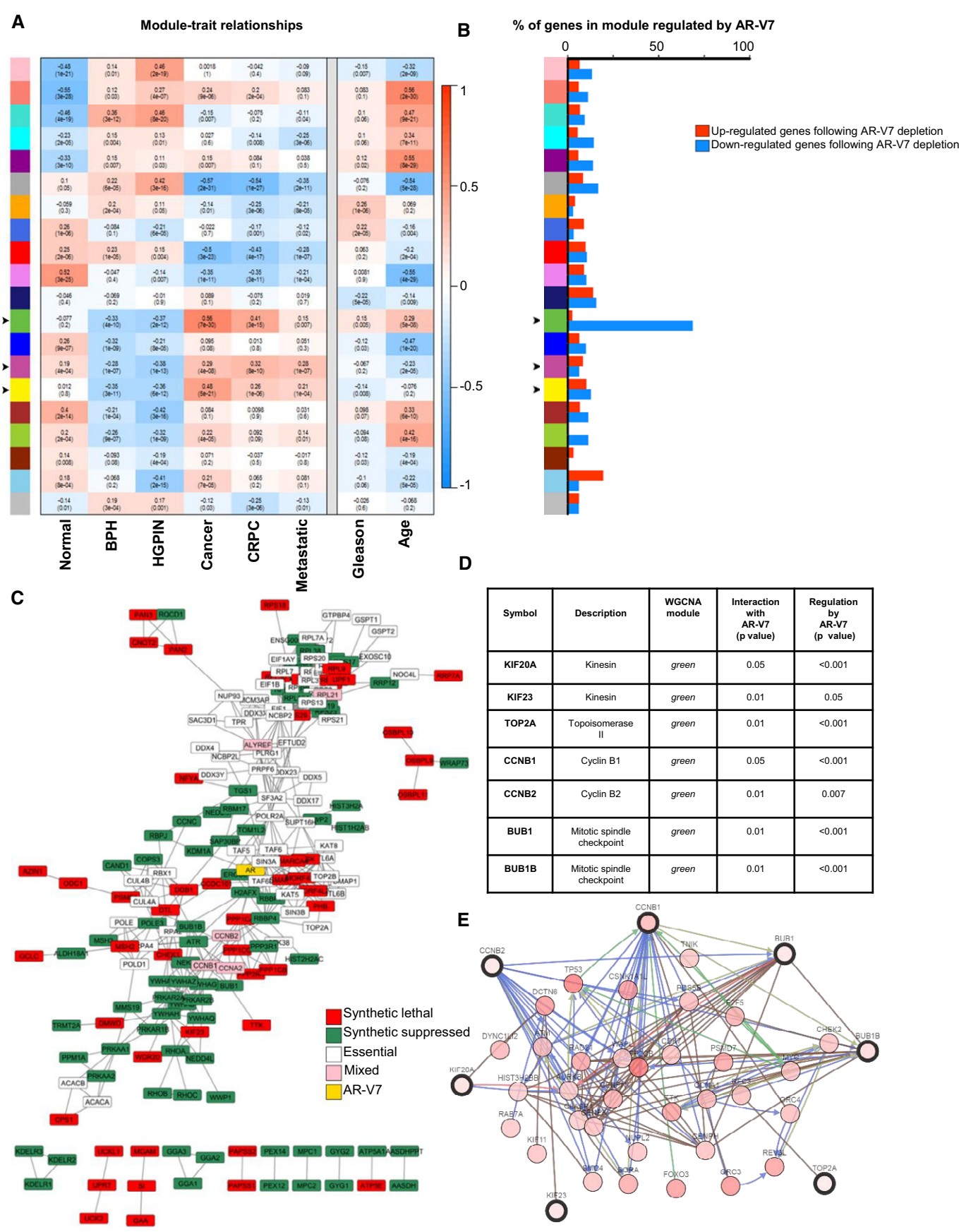

**Figure 1.**

**Figure 1.  Multifaceted system-level analyses identify seven prostate cancer hub genes.**

A   Module–trait relationships were established by WGCNA using eight independent microarray analyses comprising 375 human prostate samples. Gene modules (y-axis) are denoted by an arbitrary color name. Bins show the Pearson correlation value between gene expression levels of each module within the noted phenotype/disease stage (x-axis) and P-values. A value of 1 (red) quantifies the strongest positive correlation (genes are upregulated), −1 (blue) the strongest negative correlation (genes are downregulated), and 0 (white) no correlation. Arrows indicate those modules whose genes were positively associated with PC.

B   Microarray analysis was performed following doxycycline-regulated specific AR-V7 depletion (using a tet-pLKO backbone) in 22Rv1 PC cells compared to doxycycline-treated shGFP controls. The genes that were significantly regulated by shAR-V7 (in either direction, P-value < 0.05) were distributed among the gene modules defined by WGCNA in panel (A). Upregulated genes (red) are those in which expression increased following AR-V7 depletion, and conversely, downregulated genes (blue) are those that decreased following AR-V7 depletion. Arrows indicate those modules whose genes were positively associated with PC.

C   AR-V7 human functional *interactome* was generated using SGA screening in the yeast *Schizosaccharomyces pombe*, combined with STRING data to map protein–protein interactions, followed by the identification of the human orthologs. The colors denote the different types of genetic interactions: Red are genes that when deleted in yeast and crossed with AR-V7-expressing yeast suppressed growth, while green denotes genes that when deleted enhanced growth. White designates yeast essential genes (i.e., genes that are critical for yeast survival and thus could not be present in the yeast deletion library), but were incorporated into the network based on the criteria that they are known (based on literature) to physically interact with at least two of the red or green genes. Pink-colored genes are a combination of essential and non-essential genes identifying the same human protein.

D   Table summarizing the seven PC hub genes identified by the system-level analyses.

E   Network interactions of the seven genes with the 50 most frequently altered neighbor genes were mapped using cbioportal.org. The types of gene-to-gene interactions are as follows: controls state change (blue), controls expression (green), and in complex with (brown).

Source data are available online for this figure.

expression of AR-V7 altered the strain's growth rate ("fitness"). Gene hits therefore encode proteins that are functionally linked with AR-V7, as their deletion affected yeast growth only when AR-V7 was expressed (induced conditions), but not in the non-induced conditions. The human orthologs for gene hits were then analyzed with a protein–protein interaction network using STRING (Fig 1C). Gene ontology analysis of the AR-V7 *interactome* identified several distinct biological processes, including cell cycle regulation (Appendix Fig S5).

Integrating the data from the AR-V7 *interactome* with the AR-V7-regulated *green* module could reveal strongly disease-relevant candidates for AR-V7 feedback regulation. In this way, we identified seven genes, present in the WGCNA *green* module (Fig 1A; and thus, associated with and upregulated in PC progression), which were upregulated by AR-V7 (Fig 1B) and feedback to functionally interact with AR-V7 (Fig 1C). This seven-gene set (Fig 1D) was composed of kinesin family member 20A (*KIF20A*), kinesin family member 23 (*KIF23*), topoisomerase DNA II alpha (*TOP2A*), cyclin B1 (*CCNB1*), cyclin B2 (*CCNB2*), BUB1 mitotic checkpoint serine/threonine kinase (*BUB1*), and BUB1 mitotic checkpoint serine/threonine kinase B (*BUB1B*). This seven-gene set comprises a highly interconnected network (Fig 1E), and pathway analysis revealed a strong role in cell cycle (Appendix Fig S6).

To validate these findings, we assessed the expression of the seven genes in an independent collection of human CRPC specimens. Because these seven genes were regulated by AR-V7 in PC cells (Fig 1B), we examined whether they were co-expressed with AR-V7 in an independent gene expression profiling array dataset of human CRPC bone metastatic specimens. The human CRPC bone metastases were grouped based on the relative levels of AR-Vs expressed, mainly AR-V7 (Hörnberg *et al*, 2011; Data ref: Hörnberg *et al*, 2011). The expression of six of the seven members of the gene set were elevated in the human CRPC bone metastases with highest AR-V7 expression compared to the specimens with the lowest relative levels of AR-V7 (Fig 2A). In contrast, the expression of the members of the gene set was not associated with levels of full-length AR (Appendix Fig S7A).

In another independent PC patient dataset, we found through pairwise comparisons that the expression of the seven genes was highly correlated with each other (Appendix Fig S7B). This finding was consistent with the seven genes clustering together in the same WGCNA gene module (*green*), since modules were constructed based on correlation of gene expression. As further indication of the specificity of these associations in human PC, the expression levels of closely related genes, e.g., *BUB3* and *KIF20B*, were not correlated with any of the seven genes in PC (Appendix Fig S7B).

Analysis of an independent RNA-seq patient dataset (TCGA) revealed that the expression of the seven genes was associated with well-established adverse prognostic indicators, including Gleason score (Appendix Fig S8A), T clinical staging category (Appendix Fig S8B), and MRI evidence of extraprostatic extension (Appendix Fig S8C). The strong association between the expression levels of the seven genes and the tumor Gleason score in the RNA-seq patient data was in agreement with the WGCNA analysis, showing a significant positive correlation of the expression levels of the genes in the *green* module with Gleason score (Fig 1A).

In addition, patients whose tumors had high expression (z-score threshold ≤ 1.96) of all members of the seven-gene set exhibited significantly decreased disease-free survival (DFS) and lower overall survival in two distinct datasets compared to those patients with lower expression levels of the gene set (Fig 2B). Interestingly, despite a well-established role for these genes in cell cycle/mitotic regulation, the gene set had no association with survival metrics for a number of other major types of cancers (Appendix Fig S9), supporting a PC-specific role of this gene set.

We examined whether full-length AR also regulated the seven-gene set. Androgen stimulation of the androgen-dependent cell line LNCaP and the CRPC cell lines 22Rv1 and C4-2B did not increase the expression levels of any of the seven genes (Appendix Fig S10). In contrast, expression of the established AR-regulated gene *FKBP5* was substantially increased (Appendix Fig S10). Moreover, pairwise comparisons showed that the expression levels of the seven genes were not associated with the expression levels of AR in the TCGA Prostate Adenocarcinoma provisional patient dataset (Appendix Fig S11). These data demonstrate that ligand-activated full-length AR does not regulate the expression of the seven genes.

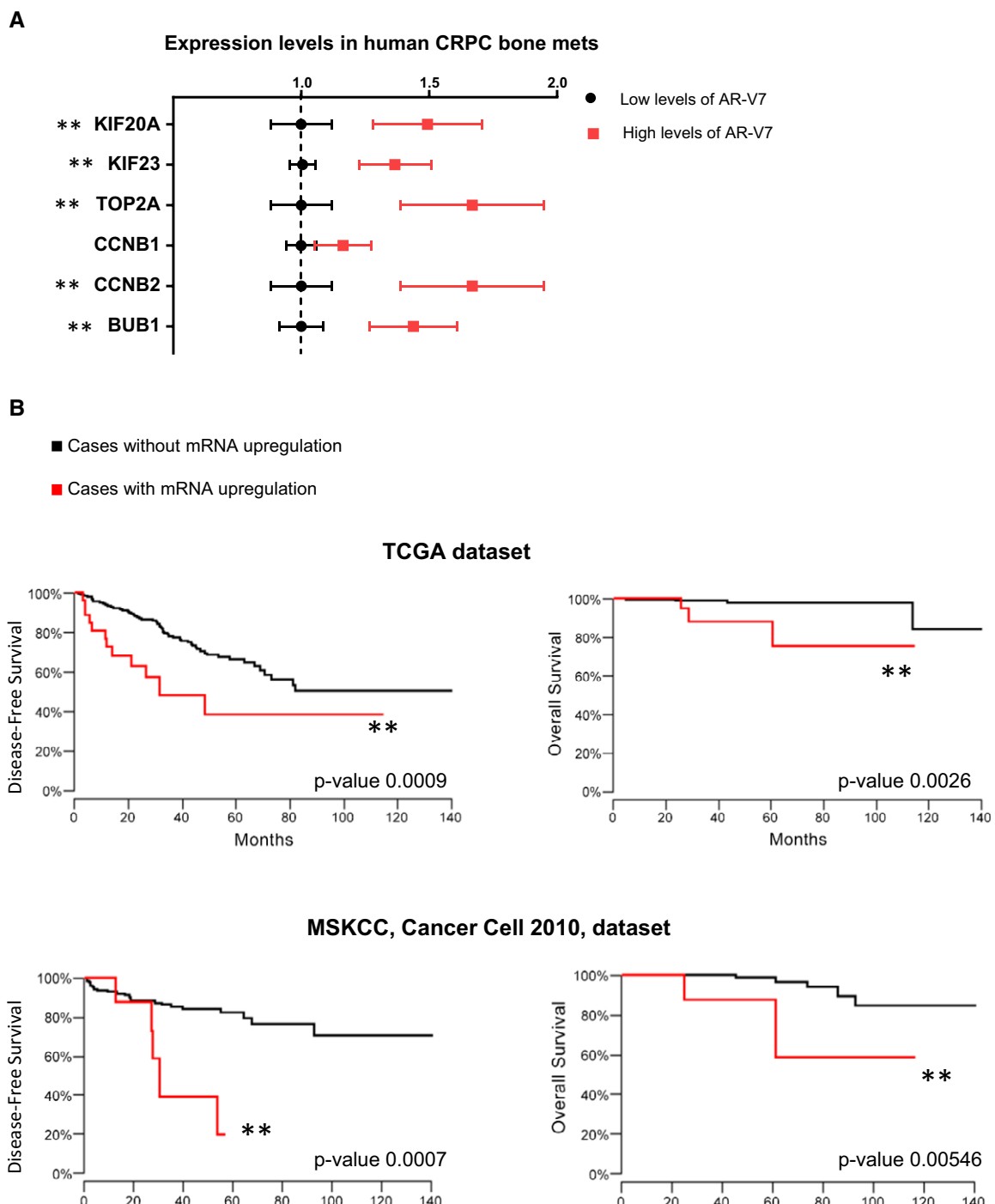

**Figure 2. Elevated expression of the seven-gene set is associated with higher levels of AR-V7 and serves as a prognostic biomarker for disease-free survival (DFS) and chances of death in PC patients.**

A   Hörnberg *et al* (2011) gene expression profiling array data were analyzed to determine the expression levels of the seven genes in human CRPC bone metastases, grouped by their relative levels of AR variants, mainly AR-V7 [high levels of AR-V7 (top quartile) or low levels of AR-V7 (quartiles 1–3)]. Data are plotted as the mean ± s.e.m. Non-parametric Mann–Whitney test was performed (two-tailed). Note that BUB1B expression was not measured in these microarrays. **P-value < 0.05. N (AR-V7 low) = 20; N (AR-V7 high) = 10.

B   The Kaplan–Meier curves for disease-free survival (DFS) and overall survival were built using the TCGA Prostate Adenocarcinoma dataset (465 samples; upper graphs). Log rank tests were performed. The black curves denote cases with normal expression of the gene set, and red represents cases where the mRNA levels of the seven genes were upregulated (z-score threshold ≤ 1.96). For DFS, *P*-value = 0.0009; for death, *P*-value = 0.0026. An independent dataset was analyzed (Prostate Adenocarcinoma MSKCC, Cancer Cell 2010, 123 samples; lower graphs). The black curves denote cases with normal expression of the gene set, and red represents cases where the mRNA levels of at least five genes of the gene set were upregulated (z-score threshold ≤ 1.96). For DFS, *P*-value = 0.0007; for death, *P*-value = 0.00546.

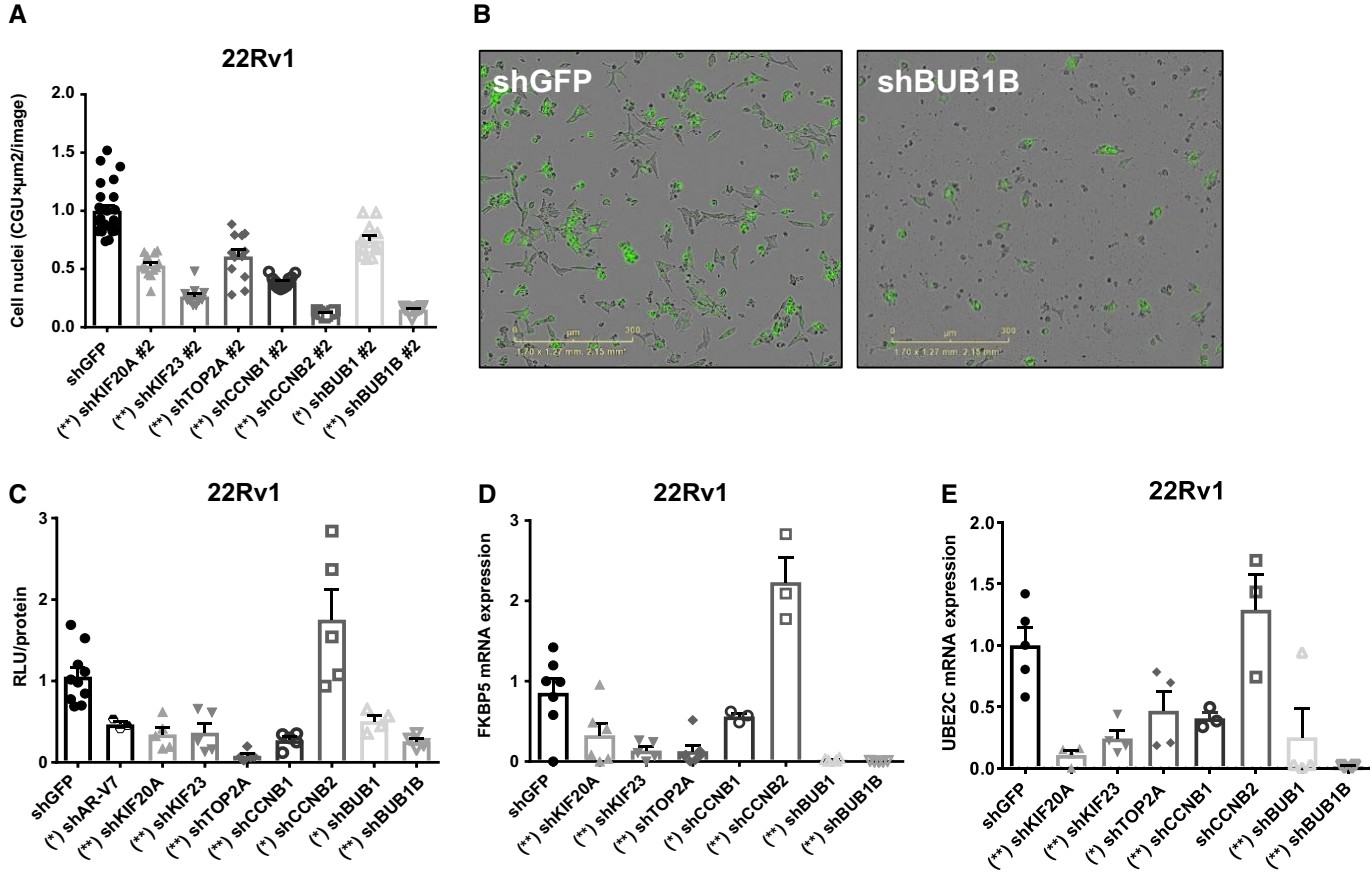

**Figure 3. Depletion of the expression of members of the seven-gene set reduces CRPC cell proliferation and AR ligand-independent transcriptional activity.**

A    Cell proliferation was examined in the CRPC cell line 22Rv1 following individual depletion of mRNAs for the seven genes or shGFP controls, using shRNA against the coding region for each gene (shRNA #2). Cell number was measured using a non-perturbing nuclear restricted dye and quantified after 72 h using Incucyte Zoom System. Data shown are mean ± s.e.m. of eight to 12 replicates normalized to their shGFP control. Kruskal–Wallis test (P-value < 0.0001, two-tailed) and Dunn's multiple comparisons test were performed. *P-value < 0.05, **P-value < 0.001.

B    Representative images of 22Rv1 stably depleted of BUB1B or control (shGFP) are shown.

C    22Rv1 stably depleted of each of the seven genes were transfected with a dual-plasmid luciferase reporter system which quantifies AR activity and basal transcription. The assay was conducted in 2% CSS to measure AR ligand-independent transcriptional activity. Data represent two independent experiments performed in triplicate, showing the mean ± s.e.m., and normalized to their shGFP controls. Kruskal–Wallis test (P-value < 0.0001, two-tailed) and Dunn's multiple comparisons test were performed. *P-value < 0.05, **P-value < 0.001.

D, E    The expression of FKBP5 and UBE2C determined by RT–qPCR analysis and normalized to GAPDH mRNA levels was examined in 22Rv1 cells stably expressing shRNA for each of the seven genes. Cells were cultured in 2% CSS. Data represent two independent experiments performed in duplicate or triplicate, showing the mean ± s.e.m., and normalized to their shGFP controls. Kruskal–Wallis test (P-value < 0.0001, two-tailed) and Dunn's multiple comparisons test were performed. *P-value < 0.05, **P-value < 0.001.

Source data are available online for this figure.

Individual depletion of the expression of the seven genes, using two distinct shRNA constructs for each gene, in the human CRPC cell lines 22Rv1 (Fig 3A and B, Appendix Fig S12A) and C4-2B (Appendix Fig S12B) decreased cell proliferation. Knockdown efficiency for each mRNA ranged from 67 to 99% (Appendix Fig S13). Since the members of the gene set were not only regulated by AR-V7 (Fig 1B), but also exhibited functional interactions with AR-V7 (Fig 1C), we investigated whether these genes modified AR-V7 transcriptional activity by reporter gene assays. Experiments were performed in 22Rv1 in which AR activity in the absence of androgen is driven by ligand-independent AR-Vs (Dehm et al, 2008; Guo et al, 2009). Depleting the expression of six of the seven genes decreased ligand-independent AR transcriptional activity (Fig 3C). Similarly,

the expression of well-known AR-V7 target genes, FKBP5 (Fig 3D) and UBE2C (Fig 3E), were significantly reduced upon knockdown of six of the seven genes in the absence of androgens. Depleting CCNB2 decreased CRPC proliferation but, in contrast to the effects of depleting the other six genes, depletion of CCNB2 elevated AR activity based on luciferase assays and regulation of FKBP5 (Fig 3C and D). No significant effect on UBE2C mRNA levels was observed following depletion of CCNB2 expression (Fig 3E). Thus, six members of the seven-gene set, which is regulated by AR-V7 and present in the AR-V7 interactome (Fig 1C), reciprocally enhanced ligand-independent AR activity in PC cells expressing AR-V7. To begin to understand the interrelationships between the seven genes and AR-V7, we explored the effects of depleting one member of each

subgroup (kinesins, cyclins, and mitotic checkpoints) of the gene set on AR-V7 levels. We found that individual depletion of KIF20A, TOP2A, and BUB1B decreased AR-V7 mRNA levels, while depletion of CCNB2 increased AR-V7 mRNA levels (Appendix Fig S14A). We also examined whether any of these members regulated full-length AR. We found that individual depletion of KIF20A and TOP2A reduced, while depletion of CCNB2 increased full-length AR mRNA levels (Appendix Fig S14B). Depletion of BUB1B had no effect on full-length AR levels. Together, these findings are consistent with the effects of depleting the expression of these genes on ligand-independent AR transcriptional activity (Fig 3C–E).

The seven-gene set may contain attractive therapeutic targets because these genes participate in interconnected cellular pathways (Fig 1A and E) and act upstream (Figs 1C and 3C–E) and downstream of AR-V7 signaling (Fig 1B). To test whether inhibition of this network decreased CRPC cell proliferation, we used doxorubicin (DOX), which inhibits TOP2A (Tacar *et al*, 2013), and N9-isopropylolomoucine (N-9), which targets CCNB1/CDK1 (Havlicek *et al*, 1997). Because of extensive pathway interactions (Appendix Fig S15), these drugs may also inhibit the activity and/or levels of CCNB2, BUB1, and BUB1B. The CRPC cell line 22Rv1 was treated with the compounds at various concentrations individually or in combination. The normalized isobologram and combination index (CI) were built and determined using *Compusyn* software. Nanomolar concentrations of the two drugs, DOX and N-9, exhibited synergistic (CI < 1) antiproliferative effects over a range of combinations (Fig 4A). We utilized the combination that had the lowest CI on a panel of different prostate cell lines (CI = 0.45). While the single agents, DOX or N-9, or the combination of the two compounds had no significant effect on the proliferation of the non-tumorigenic prostate epithelial cell line RWPE-1, or the AR-null human PC cell line PC3, the combination of both compounds synergistically inhibited the proliferation of the two CRPC cell lines 22Rv1 and C4-2B (Fig 4B and C). The two compounds also inhibited proliferation of AR-expressing androgen-dependent LNCaP cells; however, the effect was less pronounced than that observed in CRPC cells (Fig 4B). Together, the data indicate that CRPC cells exhibited a particular dependency on these seven genes for growth and survival.

## Discussion

Integrative approaches, such as those used here, transform one-dimensional cancer signatures into multidimensional networks of connecting modules (Rhodes & Chinnaiyan, 2005), which can facilitate more optimal therapeutic strategies. We identified a novel AR-V7-related gene set with prognostic and therapeutic value for PC using an integrated and unbiased data mining and experimental strategy (Fig 5), which could be readily applied to other cancer types. As just one example, this approach could be adopted for cancers that are also driven by transcription factors, such as c-Myc, KIT, and ER. Our approach included meta-analyses of gene expression profiles from human prostate tumors to derive gene modules, whose expression coincides across disease states. These modules were integrated with data obtained from human PC cells that identified AR-V7-regulated genes and with data from an AR-V7 functional network, constructed through a powerful model genetic system. This multifaceted approach, which does not use any filtering or *a priori* assumptions, resulted in the identification of disease-relevant genes that were regulated by AR-V7 and that reciprocally enhanced AR ligand-independent activity. We performed extensive intervalidation with independent patient datasets and extended findings using cell-based experimentation.

We performed a meta-analysis of microarray data on clinical PC samples, including 375 samples from eight different datasets (obtained from the same type of array so gene expression measurements could be directly compared) and encompassing six different phenotypes/disease stages. The large number of samples provided robustness to the module definition, as well as power in the ability to identify relevant modules. The gene members of the *green* module had expression levels significantly associated with and upregulated upon cancer onset and progression to CRPC, as well as Gleason score. This *green* module contained 60 genes (nearly 75%) regulated by AR-V7. Notably, AR variants regulating all the 60 genes was confirmed by RNA-seq data of He *et al* (2018). However, only a few of the genes in the *green* module (6%) are full-length AR-regulated genes.

Ligand-activated AR is well recognized as a regulator of the cyclin D-RB axis in prostate cancer (reviewed in Balk & Knudsen, 2008). However, our findings suggest that AR variants, in particular AR-V7, may be intricately related to G2-M phase cellular dynamics, consistent with Hu *et al* (2012), who demonstrated AR-V7 regulation of several genes involved in mitosis. An important implication is that the seven genes represent a vulnerability, especially but not exclusively, for AR-V7-driven CRPC and could provide possible approaches for overcoming androgen deprivation therapy and taxane resistance in CRPC patients.

---

**Figure 4.  Combined pharmacologic inhibition of TOP2A and CCNB1 synergistically inhibits CRPC cell proliferation.**

A  The CRPC cell line 22Rv1 was cultured in 2% CSS media and treated for 72 h with vehicle (DMSO), doxorubicin (DOX), N9-isopropylolomoucine (N-9), or the combination of DOX and N-9 at different concentrations. Cell confluence was monitored using Incucyte Zoom System, and the experiments were done with eight replicates each. The data were analyzed using *Compusyn* software, and a normalized isobologram was built. The table shows the combination index (CI) for the different drug combinations. CI = 1 represents additivity, CI < 1 synergism, and CI > 1 antagonistic effects.

B  The non-tumorigenic prostate epithelial cell line RWPE-1, the AR-null PC cell line PC-3, the androgen-dependent cell line LNCaP, and the CRPC cell lines C4-2B and 22Rv1 were treated for 72 h with vehicle (DMSO), DOX [100 ng/ml (184 nM)], N-9 [200 ng/ml (613 nM)], or the combination of DOX [100 ng/ml (184 nM)] and N-9 [200 ng/ml (613 nM)]. C4-2B and 22Rv1 cells were kept in 10% CSS media, and the other cell lines were kept in 10% FBS. Cell confluence was monitored using the Incucyte Zoom System. Data represent two independent experiments, with four to six replicates each, showing the mean ± s.e.m., and normalized to vehicle controls. Kruskal–Wallis test, (*P*-value < 0.0001, two-tailed) and Dunn's multiple comparisons test were performed. *\**P*-value < 0.05, \*\**P*-value < 0.01, \*\*\**P*-value < 0.001.

C  The non-tumorigenic prostate cell line RWPE-1 and the CRPC cell line 22Rv1 were treated for 72 h with vehicle (DMSO) or the combination of DOX and N-9 at 100 ng/ml and N-9 200 ng/ml, respectively. Cell confluence was monitored using the Incucyte Zoom System and representative images are shown.

Source data are available online for this figure.

**A**

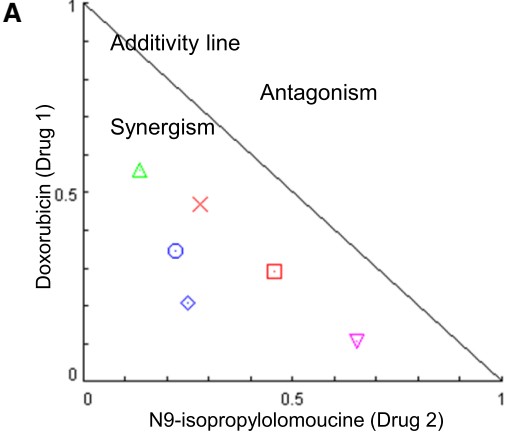

| Symbol | Dose DOX (ng/mL) | Dose N-9 (ng/mL) | Combination Index (CI) |
|--------|------------------|------------------|------------------------|
| ⊙ | 100.0 | 100.0 | 0.56669 |
| ▢ | 100.0 | 250.0 | 0.75105 |
| △ | 250.0 | 100.0 | 0.69826 |
| ▽ | 50.0 | 500.0 | 0.76248 |
| ◇ | 100.0 | 200.0 | 0.45796 |
| ✕ | 250.0 | 250.0 | 0.74909 |

**B**

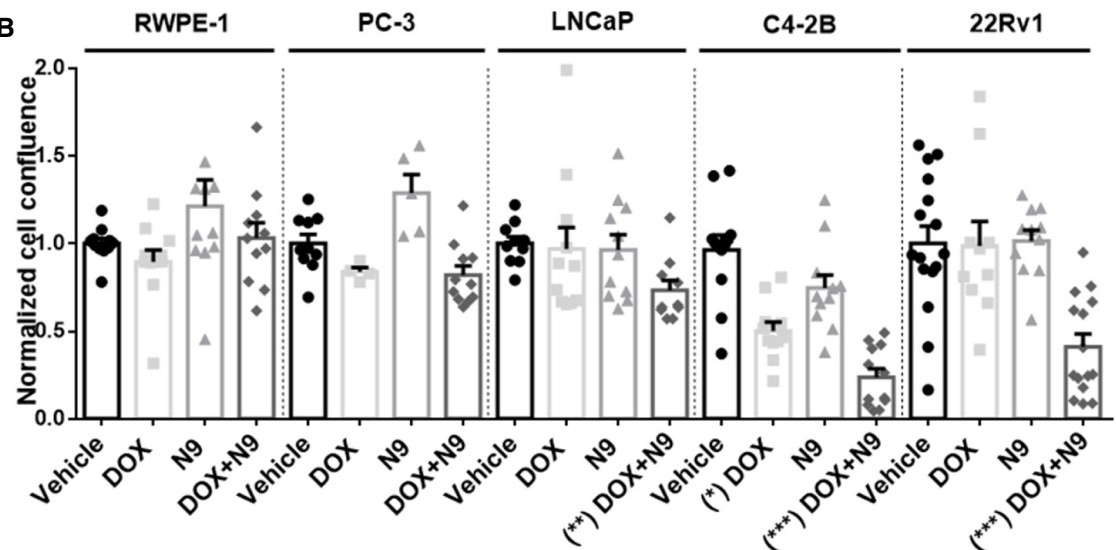

**C**

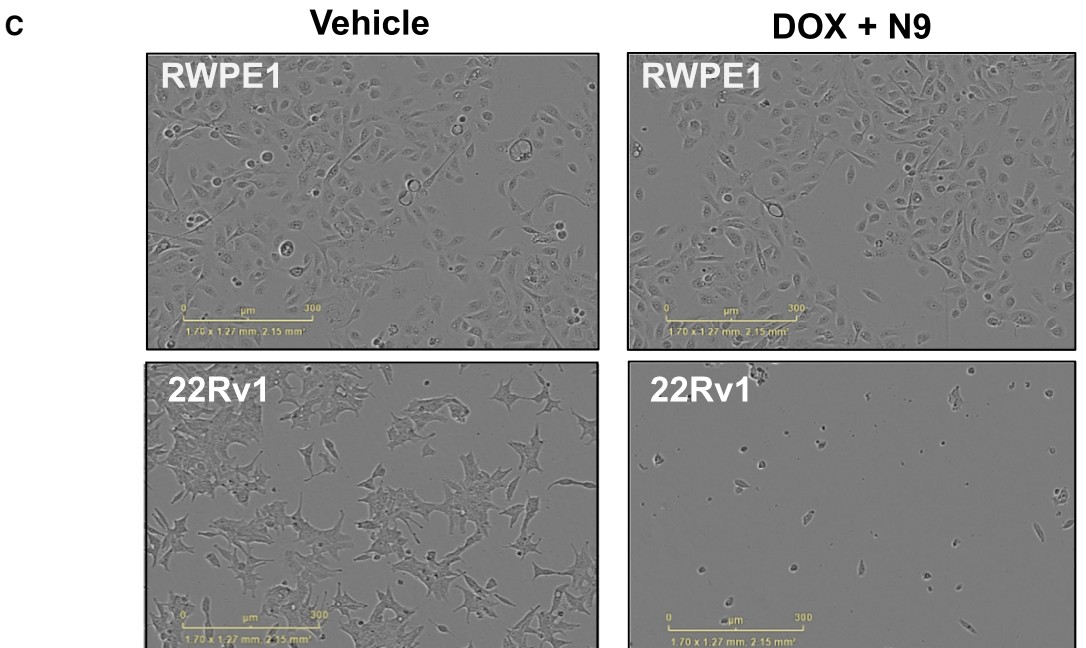

Figure 4.

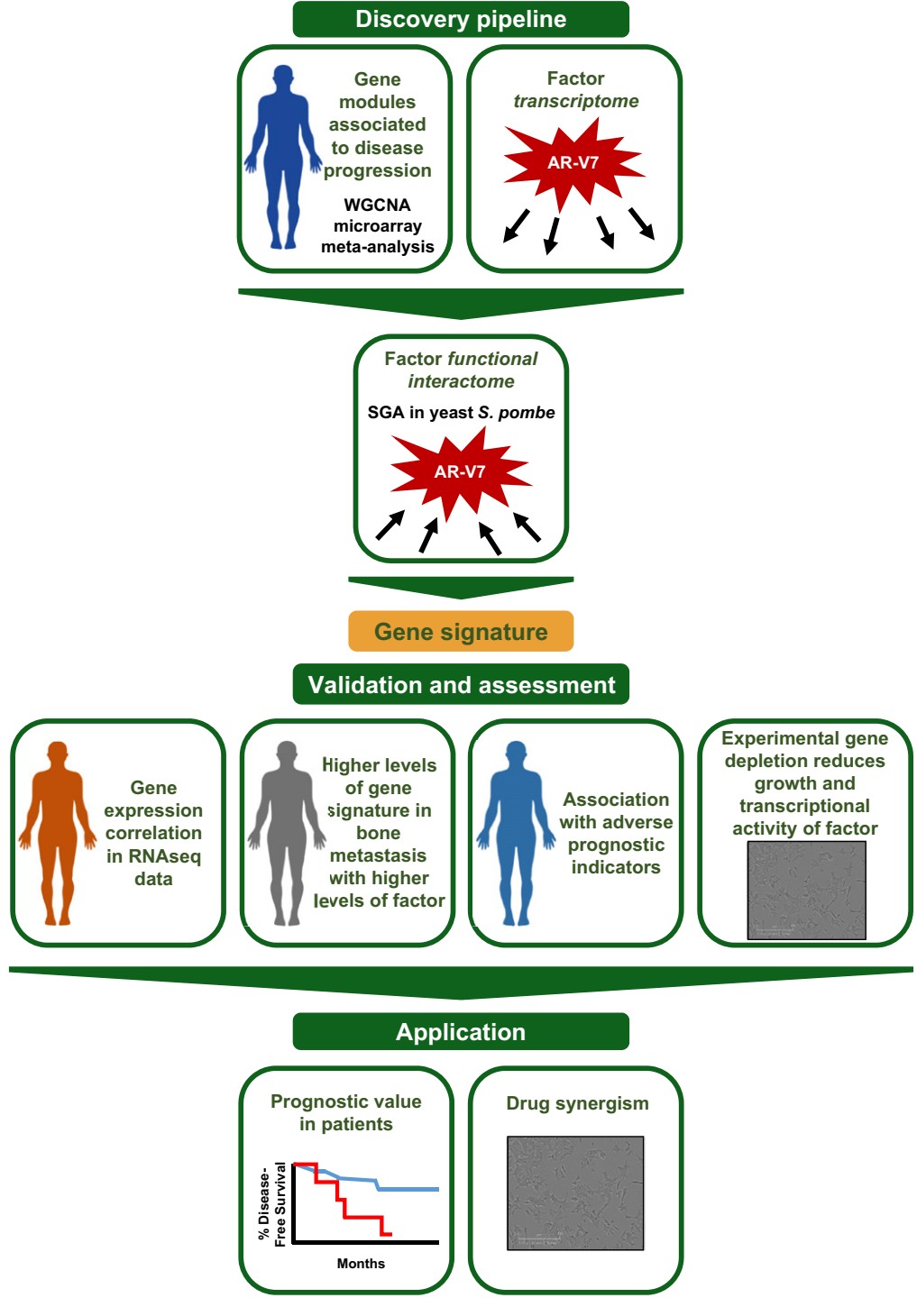

**Figure 5.   Schematic of pipeline developed, validated, and applied.**

There is a critical need to identify gene signatures that robustly predict PC aggressiveness and that can inform active surveillance disease management decisions (Cooperberg & Carroll, 2015). We found that patients with tumors exhibiting higher expression levels of the seven genes had an elevated risk of relapse after primary therapy and a greater risk of death. Thus, this seven-gene set has the potential for use in stratifying patients and guiding treatment according to clinical risk. Despite having established roles in cell cycle and cell division, which are hallmarks of all cancers, the seven-gene set did not predict patient survival metrics in other types of cancer from TCGA dataset cohorts. This finding supports a PC-specific role of this gene set, especially for cells that depend on

AR-V7 signaling. In fact, six out of the seven genes encoded proteins that enhanced AR ligand-independent activity in cell-based assays. This finding may help to explain why the seven genes are selectively associated with PC progression. Moreover, another cell cycle progression (CCP) signature has shown prognostic value in PC patients (Cuzick et al, 2011), supporting the importance of cell cycle genes in predicting PC patient outcome. However, the 31-gene CCP signature contains only three of the seven genes identified here, suggesting that the seven-gene set is distinct. The seven-gene set was derived using a strong biological rationale and a systems-biology approach and is a smaller signature, easier to apply and obtain significant predictive and survival metrics. Lastly, our signature also provided the rationale for a promising combination therapy to be explored for castration-resistant prostate cancer.

Since full-length AR is more readily targeted therapeutically than AR-V7, this work focused on AR-V7, which may promote PC therapeutic resistance and disease progression (reviewed in Luo et al, 2017). Most PC patient tissue and cell lines that express AR-V7 also express full-length AR (Guo et al, 2009; Hörnberg et al, 2011), and as AR-V7 can heterodimerize with full-length AR (Xu et al, 2015), any actions of AR-V7 likely occur in the context of full-length AR. The extent to which the full-length AR transcriptome overlaps with AR-V7 (or heterodimers of full-length AR and AR variants) is not fully understood (Watson et al, 2010; Hu et al, 2012; Cao et al, 2014; Xu et al, 2015). However, we found that ligand-activated full-length AR did not regulate the expression of any of the seven genes. Further, full-length AR mRNA levels did not correlate with the expression of the gene set in PC patients.

The seven genes we identified could interact with and promote AR-V7 transcriptional activity in various ways, and it is likely that they also participate in full-length AR networks. In fact, two members of the seven-gene set are known to regulate or enhance full-length AR activity (Chen et al, 2006; Yu et al, 2014; Schaefer-Klein et al, 2015). Li et al, 2015 showed that TOP2A inhibition reduces full-length AR and AR-V7 transcriptional activity, through decreased AR recruitment to target gene promoters and reduced nuclear localization. In addition, Chen et al (2006) showed that CCNB1/CDK1 phosphorylates the AR amino-terminus, stabilizes, and increases the transcriptional activity of full-length AR. We found that at least two members of the gene set (including TOP2A) reduced full-length AR mRNA levels upon depletion. Moreover, depletion of KIF20A as well as BUB1B decreased AR-V7 mRNA levels. CCNB2 was the one of the seven genes that upon knockdown increased ligand-independent AR activity and increased AR-V7 and full-length AR mRNA levels. Because depletion of the seven genes decreased proliferation but did not uniformly decrease ligand-independent AR transcriptional activity (as shown for CCNB2 depletion), the effects on AR activity are not likely to be secondary to reduced cell proliferation. Since the seven genes are interrelated and highly connected, it is possible that depletion of CCNB2 increased the expression of select members of the gene set in a compensatory way, which could drive the increase in AR ligand-independent transcriptional activity as well as AR-V7 and full-length AR mRNA levels. Thus, the seven identified genes, while not being regulated by full-length AR, participate at least in some settings in enhancing full-length AR. Indeed, as discussed below, LNCaP and C4-2B cells, which

are not thought to be driven by AR variants, were growth inhibited by the combination of nanomolar doses of doxorubicin and N-9. These findings support the importance of the seven-gene signature in a broader PC context beyond AR-V7-driven tumors.

Because the seven genes belong to highly interconnected pathways and networks that control each other's expression and/or activities, there is a strong likelihood that inhibition of any two of these genes would provide significantly enhanced antitumor effects. Indeed, we used two known chemotherapeutic agents as a proof of principle: doxorubicin (targeting the activity of TOP2A) and N9-isopropylolomoucine (N-9; targeting CCNB1/CDK1 activity, and indirectly affecting three other genes). The drugs, used within the nanomolar range, provided synergistic suppression of CRPC growth in 22Rv1 cells, which express AR-V7 (Peacock et al, 2012) and C4-2B, which are also highly reliant on AR signaling (Liu et al, 2014). In contrast, the two compounds had no effect on non-tumorigenic or AR-null cells, and affected the androgen-dependent LNCaP cell line to a much lesser extent. The effects of the drug combination on LNCaP and C4-2B, which do not express AR-V7, could likely be due to effects on full-length AR. Indeed, DOX and N-9 target TOP2A and CCNB1, respectively. As discussed above, we showed here that the depletion of TOP2A decreased full-length AR mRNA levels, and Li et al (2015) and Chen et al (2006) reported that TOP2A and CCNB1 enhance full-length AR transcriptional activity. CRPC cells may possess a unique dependency on these seven genes for growth and survival since these genes are not only targets of AR-V7 but also enhance ligand-independent AR activity.

In summary, we developed and used an integrative and unbiased data mining and experimental strategy to define a new AR-V7-related gene set with prognostic and therapeutic value for PC. These findings support future *in vivo* and possibly clinical studies in which combinations of these seven gene products are inhibited in PC. Additionally, this seven-gene set should be explored in prospective studies of PC to determine their prognostic capacity in different clinical risk settings.

# Materials and Methods

### Microarray dataset preprocessing

Microarray datasets were downloaded from the Gene Expression Omnibus (GEO, National Center for Biotechnology Information) or ArrayExpress (European Bioinformatics Institute). Microarrays on PC samples were selected if they used the Affymetrix Human Genome U133 plus 2.0 array and had clinical metadata for each sample (Varambally et al, 2005; Data ref: Varambally et al, 2005; Traka et al, 2008; Data ref: Traka et al, 2008; Arredouani et al, 2009; Data ref: Arredouani et al, 2014; Satake et al, 2010; Data ref: Satake et al, 2013; Jia et al, 2011; Data ref: Wang, 2009; Vaarala et al, 2012 ; Data ref: Vaarala et al, 2012; Rands et al, 2013; Data ref: Roth, 2008; Mortensen et al, 2015; Data ref: Mortensen & Dyrskjøt, 2015) (Appendix Fig S1A and B). In total, 375 samples met our inclusion criteria. These represent eight studies performed at different institutes. The arrays were assigned to one of the following groups: normal, benign hyperplasia, high-grade prostatic

intraepithelial neoplasia (basement membrane intact; HGPIN), or cancerous. The cancerous group contained nested subsets for CRPC and metastatic samples.

Microarray data were read into R and preprocessed using the "Affy" package. Preprocessing was performed as previously described (Chandran *et al*, 2016). Briefly, the "expresso" function was used to perform MAS5 preprocessing on each array. The correlation of gene expression between samples was calculated, and samples with mean correlations more than two to three standard deviations below average were excluded. Filtered samples were combined, annotated, and quantile normalized. Clinically relevant metadata were constructed from sample annotations.

## Construction of prostate cancer gene co-expression networks

The weighted gene co-expression network analysis (WGCNA) package was used to construct consensus modules containing highly connected nodes present across different PC datasets (Langfelder & Horvath, 2008). Modules were constructed with a minimum module size of 30 genes, and highly similar modules were combined using a dissimilarity threshold of 0.25. The Pearson correlation was first calculated between gene pairs. A weighting parameter, $\beta$, was applied to the correlation matrix, with $\beta$ satisfying scale-free topology criteria (Chandran *et al*, 2016). The weighted correlation matrix was used to calculate a topological overlap matrix and node dissimilarity. Genes were hierarchically clustered using the distance measure, and dynamic tree-cutting algorithm was used to define modules (Zhang & Horvath, 2005). Batch effect was minimized by using only U133plus2 arrays and removing outlier samples. All genes probed on this platform were used for the analysis. The resulting modules represent sets of highly connected nodes across the PC datasets. The first principle component of each module was correlated with the clinical data to identify module–disease state relationships. Gene ontology analysis was performed using the "GOenrichmentAnalysis" function of the "WGCNA" R package.

## Cell culture and chemical reagents

The human PC cell lines LNCaP, 22Rv1, PC-3, and RWPE-1 were obtained from American Type Culture Collection (Manassas, VA) and cultured in RPMI-1640 (Cellgro by Mediatech, Inc.), supplemented with 100 IU/ml penicillin, 100 µg/ml streptomycin, 2 mM L-glutamine (Life Technologies, Inc.), and 10% fetal bovine serum (FBS; Atlanta Biologicals) or charcoal-stripped serum (CSS) to deplete endogenous steroid hormones. The human PC cell line C4-2B was a generous gift from Dr. Conor Lynch (Moffitt Cancer Center, Tampa, FL), and cells were cultured in DMEM (Cellgro by Mediatech, Inc.), under the same conditions as mentioned above.

R1881 (methyltrienolone) was purchased from PerkinElmer Life and Analytical Sciences (Boston, MA) and used at 0.1 or 1 nM. Cells transduced with the different pLKO.1 plasmids were selected in 2.5 µg/ml of puromycin for 3 days and then maintained at 400 ng/ml of puromycin. Doxycycline was used at 100 ng/ml. Doxorubicin (DOX; from Sigma-Aldrich, D1515) was dissolved in distilled water and used at 100 ng/ml. N9-Isopropylolomoucine (N-9) was purchased from Santa Cruz (CAS 158982-15-1), dissolved in DMSO,

and used at 200 ng/ml. 22Rv1 and C4-2B cells were kept in their corresponding media with 10% CSS (androgen-depleted). For the experiments in which the effects of DOX and N-9 on cell proliferation were examined, those wells that at time zero had a cell confluence of the mean ± 1.5 times the standard deviation were excluded from analysis. The combination index (CI) was calculated using the software Compusyn, by Ting Chao-Chou and Nick Martin (http://www.combosyn.com/feature.html), based on Chou-Talalay's Combination Index Theorem (Chou & Talalay, 1984).

All cell lines were authenticated in February 2016 using STR (Genetica) and tested for mycoplasma contamination every 6 months using the Mycoplasma PCR Detection Kit (Sigma, St. Louis, MO; MP0035-1KT). All cell lines used were negative for mycoplasma, bacteria, and fungi contamination.

## Plasmids and gene depletion

The MMTV and ΔGRE/ARE luciferase plasmids were provided by Dr. Mona Nemer (University of Ottawa, Canada). The pLKO.1 shGFP and the doxycycline-induced tet-pLKO shGFP were provided by Dr. Priya Rai (University of Miami) and tet-pLKO shAR-V7 were from Dr. Yun Qiu (University of Maryland School of Medicine, Maryland). The following constructs were purchased from Sigma-Aldrich (first construct against 3′UTR, second construct against coding region): pLKO.1 shKIF20A (TRCN0000290278, TRCN0000290348), pLKO.1 shKIF23 (TRCN0000296388, TRCN0000296327), pLKO.1 shTOP2A (TRCN000049278, TRCN000049279), pLKO.1 shCCNB1 (TRCN0000293917, TRCN000045291), pLKO.1 shCCNB2 (TRCN000045193, TRCN000045197), pLKO.1 shBUB1 (TRCN0000288618, TRCN0000288618), and pLKO.1 shBUB1B (TRCN0000197142, TRCN0000194741).

## Microarray

Three independent 22Rv1 cell isolates were derived following transduction of tet-pLKO shGFP and tet-pLKO shAR-V7. Cells were grown in androgen-depleted conditions (10% CSS), plus or minus doxycycline for 1–3 days. Knockdown was evaluated via Western blot using an AR-specific antibody [rabbit polyclonal AR (N-20), Santa Cruz Biotechnology, Cat. sc-816] from a parallel protein harvest. A short-term, doxycycline-inducible knockdown system was utilized. After 48 h, RNA from the 12 samples was sent to the University of Miami Genetics Core for RNA Integrity Number (RIN) evaluation. gcRMA package was used for the analysis. Analysis was performed by examining changes in mRNA levels upon doxycycline treatment. Of the 25,293 transcripts examined, 4,273 genes exhibited significant alteration in expression following AR-V7 depletion (*P*-value < 0.05). Genes whose expression levels were significantly regulated in the shGFP control upon doxycycline treatment (628; compared to the non-induced control gene set) were removed from the list of potential AR-V7-regulated genes. This analysis identified 3,645 genes solely regulated following AR-V7 depletion (*P*-value < 0.05).

## Yeast synthetic genetic array (SGA)

To create the query strain, AR-V7 was cloned into a pENTR/D-TOPO vector (Life Technologies, Cat. K2400-20) from pcDNA3.1 AR-V7,

following the manufacturer's protocols. The following primers were used: forward primer: 5′-atggaagtgcagttagggct-3′; reverse primer: 5′-tcagggtctggtcattttgag-3′, and the genetic insertion was confirmed by sequencing (GENEWIZ). The AR-V7 gene was transferred into a destination vector using LR Gateway reactions, according to the manufacturer's protocol, to create an N-terminal HA-tagged AR-V7 under the control of the *nmt1* promoter (LR Clonase II from Life Technologies, Cat. 11791020). This expression vector was then integrated into an *h⁻leu1-32 ura4-D18 Ade6-M210 S. pombe* strain (PN572) to create an *AR-V7* query strain (*h⁻ integrated pjk148-nmt1³ˣ-HA-ARV7-nmt1ᵗᵉʳᵐ leu1-32 ura4-D18 Ade6-M210*). Strains containing AR-V7 (query strain) were grown in PMG media (Sunrise Scientific, Cat. 2060), and the expression of HA-tagged AR-V7 was induced by removing thiamine from the media after washing cells with sterile water. AR-V7 induction was confirmed by immunoblotting using an AR-V7-specific antibody (mouse monoclonal Precision antibody, Cat AG10008). Growth conditions and genetic manipulations were previously described (Moreno *et al*, 1991). The query strain was crossed to the *S. pombe* haploid deletion library (Bioneer, version 4.0 equivalent), utilizing a modified SGA procedure (Dixon *et al*, 2008). This procedure is described in detailed in Wiley *et al* (2014). Briefly, each cross was grown in four replicates under AR-V7-inducing (without thiamine) or AR-V7-non-inducing (with thiamine) conditions. Colony growth was monitored for 3 days utilizing a flatbed scanner, and plates were analyzed for "hits" (i.e., a strain with a deleted gene that when AR-V7 is expressed caused a significant growth defect or growth enhancement in comparison with the same deletion strain under non-induced conditions, *P*-value < 0.05). Essential genes were identified as hits based on the criteria that they interact with at least two primary hits in a *S. pombe* protein network (STRING, high confidence of 0.7). The identified *S. pombe* genes were converted into human orthologs using Homologene (http://www.ncbi.nih.gov/homologene; build 67), INPARANOID (http://inparanoid.sbc.su.se/cgi-bin/index.cgi), OrthoMCL (http://orthomcl.org/orthomcl/; version 5), and Pombase (www.pombase.org; build 2014-03-17-v1). Network maps were then generated in STRING at high confidence (0.7) using either experimental data (BIND, DIP, GRID, HPRD, IntAct, MINT, and PID) or experimental data and database data (Biocarta, BioCyc, GO, KEGG, and Reactome).

### Gene co-expression, Gleason score, pathologic stage, and MRI evidence of extraprostatic lesions analyses

The TCGA Prostate Adenocarcinoma provisional dataset (*n* = 499) was used. Gene co-expression analyses were performed using cbioportal.org with a Fisher's exact test, where a *P*-value < 0.05 denotes a significant association between the genes (Cerami *et al*, 2012; Gao *et al*, 2013). For the other analyses, the dataset with the clinical information was downloaded from UCSC Xena: http://xena.ucsc.edu and analyzed.

### Kaplan–Meier curves for disease-free survival (DFS) and death

The TCGA Prostate Adenocarcinoma provisional (*n* = 465) and the Prostate Adenocarcinoma MSKCC, Cancer Cell 2010 (*n* = 123) datasets were downloaded from FireBrowse (http://gdac.broadinstitute.org) and analyzed using the R package "survival". The *z*-score threshold was ≤ 1.96.

### Reporter gene assays and transfections

A dual-plasmid mouse mammary tumor virus (MMTV)-luciferase system was used in which one plasmid encodes wild-type MMTV promoter, while the control plasmid lacks androgen/glucocorticoid response elements (ΔGRE/ARE). Non-AR-driven transcriptional activity and transfection efficiency can be accounted for by utilizing the ΔGRE plasmid as a baseline control. Transfection of luciferase constructs was performed using Lipofectamine (Invitrogen Life Technologies) and PLUS reagent (Invitrogen Life Technologies), according to the manufacturer's instructions. 22Rv1 were plated at a density of $3.0 \times 10^5$ cells in 35-mm dishes 24 h before transfection. Immediately before transfection, media were replaced with unsupplemented DMEM. After a 6-h incubation period, the media were removed and cells were kept in RPMI 1640 supplemented with 2% CSS. After 48 h, cells were harvested, lysed, and assessed for luciferase activity blindly using the Promega luciferase assay kit (Promega Corp.) and a luminometer.

### Cell growth assays

Cells were plated in 96-well plates at 5,000 cells/well (for RWPE-1) or 7,500 cells/well (for 22Rv1, C4-2B, PC3), in 6–12 replicates. 22Rv1 cell lines with stable gene depletions (see Plasmids and gene depletion section) were also transfected with 2% v/v of non-perturbing nuclear restricted green fluorescent label (IncuCyte NucLight Green BacMam 3.0, Essen Bioscience). After 2 h, cells were incubated in an Incucyte Zoom (Essen Bioscience), acquiring phase (and green fluorescent images when appropriate) at 10× every 2 h. The Incucyte Zoom software was used to analyze and graph the results blindly. Each well measurement was normalized to the number of cells at the initial time and then normalized to the control (shGFP or vehicle treatment accordingly).

### RNA isolation and reverse transcriptase quantitative RT–qPCR

Total RNA was collected using TRIzol according to the manufacturer's protocol (Life Technologies) and isolated using Direct-zol RNA MiniPrep Plus (Zymo Research, Catalog number R2072). Total RNA was reverse-transcribed using a cDNA Reverse Transcription Kit (Applied Biosystems, Catalog number 4368814) as per the manufacturer's protocol. TaqMan probes from Applied Biosystems for FKBP5 (Hs01561006_m1), UBE2C (Hs00964100_g1), KIF20A (Hs00993573_m1), KIF23 (Hs00370852_m1), TOP2A (Hs01032137_m1), CCNB1 (Hs01030099_m1), CCNB2 (Hs01084593_g1), BUB1 (Hs01557695_m1), BUB1B (Hs01084828_m1), and GAPDH (Hs02786624_g1) were used.

### Statistical analysis

Data were graphed and analyzed using Prism 7 (GraphPad) and Statistica 8.0 (Statsoft). Data were tested for normality (Shapiro–Wilk test) and homogeneity of variances (Levene's test). When assumptions were met, data were tested for significance ($P < 0.05$) using a two-tailed Student's *t*-test (two groups) or analysis of variance (ANOVA; three or more groups). Otherwise,

Welch's correction or non-parametric statistical analyses were used: Mann–Whitney's test (two groups) and Kruskal–Wallis (three or more groups).

## Data availability

The microarray data that support the findings of this study are available in the following databases:

- Microarray data: Gene Expression Omnibus GSE104572 (https://www.ncbi.nlm.nih.gov/geo/query/acc.cgi?acc=GSE104572).

**Expanded View** for this article is available online.

## Acknowledgements

We are grateful to Drs. Sawsan Khuri, Tan Ince, Enrique Mesri, Irina Agoulnik, Sandra Lemmon, Priyamvada Rai, and Nagi Ayad for helpful advice; to Mr. Dimitri Nader for assistance with experiments; to Ms. Meenakkshy Manoharan for help with the manuscript preparation; and to Ms. Ann M. Greene for editing the manuscript. This research was conducted using the resources of the University of Miami Center for Computational Science (CCS) and the Onco-genomics Shared Resource of Sylvester Comprehensive Cancer Center. Research performed in this manuscript was supported by NIH Grant CA132200 (KLB), NIH predoctoral fellowship F30AG038275 (SOP), Women's Cancer Association (KLB), and by developmental and shared equipment funds from the Sylvester Comprehensive Cancer Center.

## Author contributions

FM contributed to the design of experiments, data acquisition, data analysis, and writing of the manuscript. ERB contributed to the design of experiments, data acquisition, and data analysis. VAC, MJM, NZ, LH, and SOP contributed to data acquisition. DJW contributed to data acquisition and data analysis. GD'U contributed to data analysis and provided reagents and equipment. KLB contributed to the design of experiments, data analysis, and writing of the manuscript and provided reagents and equipment.

## Conflict of interest

The authors declare that they have no conflict of interest.

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
