## [Review Process File · Molecular Systems Biology]

Identification of an oncogenic network with prognostic and therapeutic value in prostate cancer

Fiorella Magani, Eric R. Bray, Maria J. Martinez, Ning Zhao, Valeria A. Copello, Laine Heidman, Stephanie O. Peacock, David J. Wiley, Gennaro D'Urso, and Kerry L. Burnstein.

Review timeline:

First Submission:	10 th January 2018
Editorial Decision:	16 th February 2018
Author's correspondence:	26 th April 2018
Second submission:	5 th May 2018
Editorial Decision:	26 th June 2018
Revision received:	11 th July 2018
Accepted:	17 th July 2018

Editor: Maria Polychronidou.

Transaction Report:

1st Editorial Decision

16th February 2018

Thank you again for submitting your work to Molecular Systems Biology. We have now heard back from two of the three referees who agreed to evaluate your manuscript. Since their recommendations are quite similar, I prefer to make a decision now rather than further delaying the process. As you will see below, the reviewers raise substantial concerns on your work, which unfortunately preclude its publication in Molecular Systems Biology.

The reviewers acknowledge that the proposed approach for biomarker identification seems potentially interesting. However, they point out that as it stands the level of mechanistic insight provided by the study remains rather limited and the prognostic value of the proposed signature is not convincingly supported. Moreover, reviewer #2 mentions that the identified prognostic gene module, overlaps with an existing prognostic tool, which somewhat detracts from the overall novelty of the findings. Both reviewers rated the conclusiveness and the conceptual advance as "Medium/Low" and indicated that they do not support publication of the study in Molecular Systems Biology. _____

REFeree REPORTS

Reviewer #1:

Title - Systems analyses identify an oncogenic network with prostate cancer prognostic and therapeutic value;
by Magani, F. et al.

Manuscript Number - MSB-18-8202

The present study addresses the specific contribution of the AR-V7 splice variant of the androgen receptor (AR) in androgen-independent forms of prostate cancer (PC); the wild type AR is a key modulator of prostate cell growth and therefore maintenance of tumor growth under many conditions. The AR-V7 splice variant, which is a ligand-independent variant of the AR as it lacks the ligand-binding domain, has been one of the better studied AR variants that promote castration - /androgen-ablation resistant PC.

The authors identify a prostate specific 7-gene-set suggesting its prognostic and therapeutic value by using data mining techniques and experimental strategies. The authors assess the 7-gene-set using independent patient datasets. The authors attempt to generate a list of AR-V7 regulated genes by capturing gene expression profiles after knocking-down AR-V7 in a human prostate cancer cell line that has known activity of AR-V7. Using a synthetic genetic array (SGA) assay on another organism model, the authors suggest a AR-V7 functional interactome in order to find protein coding genes that are functionally linked to AR-V7 expression. The authors identified several gene modules associated with disease progression using independent publicly available microarrays datasets from patients as input for the Weighted Gene Co-expression Network Analysis (WGCNA) algorithm. By matching the genes from a WGCNA module, which is found to be associated to disease progression, with the SGA outcome, and the gene list from the knock-down experiment, the authors present 7 protein coding genes in human as candidates involved regulation by AR-V7. The authors use gene expression profiling following AR-V7 knockdown in the 22Rv1 cell line, coupled with a novel and recently described yeast genetic screen (Yeast Augmented Network analysis; YANA), and subsequent gene validation experiments to validate the module of 7 mitosis/cell cycle-related genes that they propose contributes significantly to AR-V7-driven phenotype.

Strength: The study is particularly noteworthy in that the authors attempt to address the specific contribution of the AR-V7 splice variant independent of the contribution of other forms of AR or the full-length AR (AR-FL) using the yeast YANA system. This approach has value in the context of the co-expression of AR forms in PC, and potentially could be extended to other variants.

Major Points:

1. This reviewer is quite enthusiastic about the concept of this study, and the use of novel genetic systems, but not the manuscript in its present form.
2. Critically, the specificity of the genes to AR-V7 is in question. The absence of an AR-FL control in most experiments and indeed any other meaningful control besides GFP, especially in the yeast genetic system, makes it difficult to assess the value of the genes identified using AR-V7. In YANA, expressing a full length AR-FL could equally well have pulled up these hits since they are proliferative/cell cycle genes. Note the question is not whether AR responses regulate these genes, but whether the AR-V7 variant specifically regulates these genes.
3. Further, this issue of specificity is critical to evaluate the value of the YANA genetic system as a whole, as applied to this study. Could any transcriptional activator, for e.g., c-myc, also activate the same cell cycle genes in yeast, simply because they are the most homologous to mammalian genes and c-myc is a proliferation enhancer?
4. Prior studies on 22Rv1 and AR-V7 must be outlined. Do these cells express AR-FL? Critically, what happens to the gene set if AR-FL is silenced in 22Rv1? The reviewer is aware of the history of 22Rv1, but the manuscript cannot rely on the readers knowledge to decipher this.
5. The studies conceptually have really good potential for novel discovery (like the synthetic lethality), but not as presented in this manuscript. Indeed, if the authors at least show that computationally AR-FL co-expression of these genes is differentiated from AR-V7 then there would be increased value to these studies, especially because these genes appear to stratify patient survival and specificity for PC, a point of interest in this disease, but their specific relationship to AR-V7 cannot be determined as presented.

Specific points:

1. The manuscript is short on a lot of specifics about experiments in the "Results", "Methods" or "Legends" sections, and makes evaluating the data problematic. In Figure 1C, "essential genes" in white "were put back", but what does "essential genes" mean. In Figure 1A, how was AR-V7

knocked down? Even the "Methods" does not outline this, and the reviewer infers this from personal experience! A sentence in the Results like "...knockdown using an AR-V7 shRNA in a Tet (Doxycycline) inducible pKO.1 background" or such would be useful. Similar omissions are present throughout that make reading the paper technically challenging.

2. Why 22RV1?? Is there data on AR-V7 expression in 22RV1? Specificity of the shRNA?? Data on knockdown (Western or qPCR)?? The choice of 22RV1 is critical to the experiments, so it needs to be justified. Prior studies on 22RV1 and AR-V7 must be outlined. Do these cells express AR-FL? What happens to the gene set if AR-FL is silenced in 22RV1? The reviewer is aware of the history of 22RV1, but the manuscript cannot rely on the readers knowledge to decipher this.
3. In Figure 1B, what is the full 60 gene set that is regulated by AR-V7 knockdown? A table containing this in supplements would be very useful.
4. As an aside, in Fig 1A, the grey module (#6, counting from top) and the red module (#9) are interesting in containing suppressor activities in Cancer/BPH/HGPIN.
5. In Figure 2A, the authors say "expression levels of the seven genes in human PC bone metastases"? Why only metastases? The authors should clarify what happens to "Cancer" and "CRPC" groups....the reviewer presumes this is the subdivision since nothing is clarified about the specifics of the analyses.
6. Data for yeast selection is completely lacking - metrics for growth suppression or enhancement?? Colony pictures or the like?? Western blot for successful induction of AR-V7. Etc.
7. In Figure 3A and 3B significant details are missing. Concentrations ?? For how long? Endpoint? Surprisingly, panel (F) and (G) contain abundant details, which need to be moved to earlier panels.
8. In Figure 3C, the authors say "22RV1 stably depleted of each of the seven genes were transfected with a dual plasmid luciferase reporter system which quantifies AR activity and basal transcription". Stably depleted? How do the cells survive? Is this a Doxycycline inducible? The system quantifies total AR activity. Then how is this specific to AR-V7? Are these genes generally controlled by AR?? Please explain.

Reviewer #2:

Magani et al. have used an integrative and unbiased data mining strategy and systems analyses to define a new AR-V7 related gene set with prognostic and therapeutic value for prostate cancer. In addition, in vitro experiments were carried out to support their conclusions. These findings could be the basis for preclinical validation experiments in vivo. The strategy presented in this work could be a novel strategy for finding biomarkers for complex diseases. However, the prognostic gene module identified has very significant overlap with a previously identified prognostic tool, Cell Cycle Progression (CCP) score, which performs very well for prostate cancer, as well as some other cancers. Surprisingly, the authors did not even mention this in the manuscript.

This is a well-written manuscript with significant data consistent with the conclusions drawn. However, the in vivo significance of the findings are at present not explored. There are also a number of significant points that need to be addressed:

1. Line 96-100: The authors used a specific shRNA to knockdown AR-V7 in 22RV1 cells. Experimental evidence, such as by qPCR and immunoblotting, should be provided to show that the effect is specific to the splicing variant, and not the full-length AR.
2. line 140-142: The gene expression dataset from the Homberg study is not a RNA-seq dataset.
3. line 162-164: The authors state that expression of the 7-gene signature is prognostic specifically for prostate cancer. This is not warranted as only one cohort with a couple of types of cancer was used in the analysis. Comprehensive analysis of multiple cohorts of at least major types of cancer should be conducted to assess this possibility.
4. There are very few samples (8) of CRPC included. This should be expanded.
5. "Depleting CCNB2 did not significantly reduce ligand-independent AR activity as measured in either assay". In fact, CCNB2 depletion significantly increased AR activity. What could be the mechanism for this? Does it also affect full-length AR?
6. In Fig. 3G, there appears to be at least additive effects for N9 vs N9+Dox in PC3 cells, while it is stated that there are no effects in this cell line. In addition, the combinatorial treatment should be tested in non-CRPC but AR-responsive cell lines, such as LNCaP and VCaP, to assess the specificity of the treatment for AR-V7 mediated growth.

7. Again in Figure 3g, the combinatorial treatment have similar efficacy on an AR-V7 negative cell line (C4-2B) and AR-V7 positive cell line (22Rv1). This observation actually challenges the authors' statement that the 7-gene module is specific to AR-V7.
8. The mechanisms of how the products of the seven genes identified can regulate AR-V7 activity are not explored. Is it on the expression of AR-V7, its stability, nuclear localization, etc.? Are the effects direct, or require additional factors?
9. Since most prostate cancer patient tissue as well as the cell lines that express AR-V7 also express full length AR, it would be important to determine if the 7-gene model also affects the full length AR, and/or its heterodimerization with AR-V7.

Author's correspondance

26th April 2018

We respectfully ask that you permit us to resubmit our manuscript entitled "Systems analyses identify an oncogenic network with prostate cancer prognostic and therapeutic value" for consideration in *Molecular Systems Biology*. The reviewers' comments were very helpful and we have significantly improved the manuscript in line with the critiques. As described below, we addressed the three overriding issues: 1) whether our seven gene set was specific for the androgen receptor (AR) splice variant AR-V7 or could also be regulated by full length AR; 2) possible mechanisms by which the genes regulate AR-V7 and the consequences of this regulation; and 3) overlap between our gene signature and an existing Cell-Cycle Progression (CCP) signature, which detracted from the overall impact of our findings.

1. We addressed the first point experimentally as well as through bioinformatics analyses of patient samples. We found that none of the genes in the seven gene set are regulated by full length AR as assessed in three different prostate cancer cell lines. Further, the levels of the seven genes are specifically associated with AR-V7 levels in patient samples, but not with full length AR. This work is presented in three additional supplemental figures (SF3, SF10, SF11).

2. We addressed the second point experimentally in prostate cancer cell lines. We found that several members of the seven gene set regulate AR-V7 mRNA and/or protein levels. We also demonstrate that CCNB1 (another member of the seven gene set) endogenously interacts with AR-V7. Together these results provide considerable insight into the reciprocal regulation of these critical proteins in prostate cancer progression.

3. Although the reviewer stated that our seven gene signature "has very significant overlap" with the 31 gene CCP signature, this is not correct. Only three of the seven genes are found in the 31 gene CCP signature. Further, our seven gene signature was derived using a strong biological rationale and a systems-biology approach. In contrast, the 31 gene CCP signature was derived by choosing genes already well known to regulate cell cycle. Not surprisingly the CCP gene signature is prognostic for prostate as well as other cancers. Whereas, the seven gene signature we developed is specific for survival metrics in prostate cancer. We compared the performance of our signature to the CCP, utilizing a simple cutoff threshold for two different prostate cancer datasets (TCGA and MSKCC). While our seven gene signature predicted disease-free survival and overall-survival for both datasets, the CCP signature was only able to predict disease-free survival in the TCGA dataset. A smaller signature such as the one we identified is easier to apply and obtain significant predictive and survival metrics. Lastly, our signature also provided the rationale for a promising combination therapy to be explored for castration resistant prostate cancer. We now include discussion of the CCP signature in our manuscript.

We have also addressed in the manuscript virtually all of the remaining points raised by the reviewers resulting in a total of six new figures. The results of our integrative and unbiased data mining and experimental strategy defined a new AR-V7 related gene set with prognostic and also therapeutic value for prostate cancer. This novel systems-biology approach could be readily applied to uncover new prognostic markers and therapeutic targets for other human diseases making this manuscript of interest to not only prostate cancer researchers but also more broadly to your readership.

Response to reviewers

We thank the reviewers for their helpful suggestions, which we have used to improve our revised manuscript. Below are the detailed responses to each of their questions and comments.

We have reframed the conclusions to stress that we have identified actionable downstream targets of AR-V7 that enhance overall AR signaling (and not solely AR-V7) in CRPC.

Reviewer #1:

Major Points:

1. *This reviewer is quite enthusiastic about the concept of this study, and the use of novel genetic systems, but not the manuscript in its present form.*

Critically, the specificity of the genes to AR-V7 is in question. The absence of an AR-FL control in most experiments and indeed any other meaningful control besides GFP, especially in the yeast genetic system, makes it difficult to assess the value of the genes identified using AR-V7. In YANA, expressing a full length AR-FL could equally well have pulled up these hits since they are proliferative/cell cycle genes. Note the question is not whether AR responses regulate these genes, but whether the AR-V7 variant specifically regulates these genes. Further, this issue of specificity is critical to evaluate the value of the YANA genetic system as a whole, as applied to this study. Could any transcriptional activator, for e.g., c-myc, also activate the same cell cycle genes in yeast, simply because they are the most homologous to mammalian genes and c-myc is a proliferation enhancer?

We appreciate the reviewer's overall enthusiasm and thank the reviewer for raising these important points. We have performed a number of experiments and additional data analysis to evaluate the specificity of AR-V7 vs full length AR with respect to regulating genes contained in the green module and more specifically to regulation of the seven gene set.

We first examined whether the genes in the *green* module (identified through WGCNA) contained known targets of full length AR in PC. We utilized two full length AR regulated gene signatures: one consisted of genes differentially expressed in tumor versus normal samples and enriched for AR binding sites, obtained from Pomerantz et al., 2015; and a second transcription-based full length AR activity signature from Mendiratta et al., 2009. We examined the distribution of these full length AR regulated genes across the WGCNA modules (new figure SF3). We found that no gene in the *green* module was regulated by full length AR using the full length AR signature from Mendiratta et al., and only 6% of genes in the *green* module were full length AR targets based on the Pomerantz et al. dataset. This result is in contrast to our finding that nearly 75% of *green* module genes were regulated by AR-V7 in CRPC cells. These results suggest that the *green* module is largely and selectively regulated by AR-V7, but not full length AR. Thus, we identified 60 AR-V7 regulated genes whose expression is significantly upregulated in PC, CRPC, and metastasis in the WGCNA meta-analysis of human samples.

We compared the expression levels of each member of the seven gene set with the expression levels of full length AR in two independent human datasets (Hornberg et al., 2011; and TCGA prostate cancer) (new figures SF7a and SF11). We found no correlation between the expression levels of the seven gene set members and full length AR.

In cell-based assays, we found that ligand-activated full length AR did not increase (or decrease) the expression of any of the seven genes in three different prostate cancer cell lines (new figure SF10). Thus, the *green* module, and in particular the seven gene set, are regulated by AR-V7, but not full length AR.

With regard to using full length AR in YANA, our objective was to define genes that functionally interacted with AR-V7 and that might enhance AR-V7 function in mammalian cells, but not to define interactions that were exclusive to AR-V7. We completely agree with the reviewer that many (if not all) of the YANA “hits” are also likely to functionally interact with full length AR. Using an AR variant obviated the need for ligand and also allowed us to focus on a form of AR that is strongly associated with CRPC. We did find that while the seven gene set interacted with

AR-V7, at least three of the seven encoded proteins enhanced full length AR (previous literature cited in paper and new figure SF14). In this work we have focused on genes associated with disease progression that are regulated by AR-V7, and that functionally (but not exclusively) interacted with AR-V7.

That at least some of the seven genes regulate full length AR broadens the impact of our findings and provides an explanation for why pharmacologic inhibition of this network also decreased proliferation of CRPC that lacks AR-V7 (discussed below).

With respect to the specificity of the interactions revealed by synthetic genetic array screening (for yeast growth), this approach is well-validated in the literature. In particular, Costanzo et al., 2010 (*Science*) screened 1712 genes in budding yeast, and constructed a genome-scale genetic interaction map by looking into 5.4 million gene-gene pairs for synthetic genetic interactions. These gene combinations revealed different interactions between genes as well as a substantial number of biological processes or pathways (such as protein folding, DNA replication and repair, metabolic processes, mitosis, RNA processing, etc.), showing that this method can successfully identify specific functional interactors for different gene products. As an example, Wiley et al., 2014 utilized the same model system as employed in our manuscript to build a functional *interactome* for UBA1, a gene which causes X-linked spinal muscular atrophy. YANA screening using UBA1 resulted in a set of interacting partners for UBA1 with very little overlap to the hits we obtained using AR-V7. This finding indicates that hits are specific to the gene under study and are not an artefact of the screening system.

2. Prior studies on 22Rv1 and AR-V7 must be outlined. Do these cells express AR-FL? Critically, what happens to the gene set if AR-FL is silenced in 22Rv1? The reviewer is aware of the history of 22Rv1, but the manuscript cannot rely on the readers knowledge to decipher this. The studies conceptually have really good potential for novel discovery (like the synthetic lethality), but not as presented in this manuscript. Indeed, if the authors at least show that computationally AR-FL co-expression of these genes is differentiated from AR-V7 then there would be increased value to these studies, especially because these

genes appear to stratify patient survival and specificity for PC, a point of interest in this disease, but their specific relationship to AR-V7 cannot be determined as presented.

We thank the reviewer for raising these points. We have added more detailed background information on 22Rv1 in the manuscript and outlined information on 22Rv1 from published studies (lines 98-102). As mentioned above, we have now shown, through different in vitro and bioinformatic analyses, that the green module (and the seven gene set within it) are specifically regulated by AR-V7 and not by full length AR (new figures SF3, SF7a, SF10, SF11).

Specific points:

- 1. The manuscript is short on a lot of specifics about experiments in the "Results", "Methods" or "Legends" sections, and makes evaluating the data problematic. In Figure 1C, "essential genes" in white "were put back", but what does "essential genes" mean. In Figure 1A, how was AR-V7 knocked down? Even the "Methods" does not outline this, and the reviewer infers this from personal experience! A sentence in the Results like "...knockdown using an AR-V7 shRNA in a Tet (Doxycycline) inducible plKO.1 background" or such would be useful. Similar omissions are present throughout that make reading the paper technically challenging.*

We apologize for the lack of important experimental details. We have added information and experimental details in the Results, Methods and Legends sections. We performed doxycycline-inducible knock-down of AR-V7 using a specific tet-plKO shAR-V7 construct and now show these data (SF2a and Peacock *et al.*, 2012).

In regards to YANA, we have clarified that in the interactome, white designates yeast essential genes (ie: genes that are critical for yeast survival, and thus could not be deleted and represented in the yeast deletion library), but were incorporated into the network based on the criteria that they are known (based on literature) to physically interact with at least two of the red or green genes (that were experimentally identified in the SGA screening).

2. *Why 22RV1?? Is there data on AR-V7 expression in 22RV1? Specificity of the shRNA?? Data on knockdown (Western or qPCR)?? The choice of 22RV1 is critical to the experiments, so it needs to be justified. Prior studies on 22RV1 and AR-V7 must be outlined. Do these cells express AR-FL? What happens to the gene set if AR-FL is silenced in 22RV1? The reviewer is aware of the history of 22RV1, but the manuscript cannot rely on the readers knowledge to decipher this.*

We apologize again for these omissions. We have now added more details and justifications about the cell line 22Rv1 (lines 98-102). We chose 22Rv1 cells since they contain high levels of AR-V7 and depend on AR-V7 for growth and survival (Dehm et al. 2008; Guo et al., 2009; Marcias et al., 2010). We have addressed the specificity of the shRNA for AR-V7 by showing a WB in the new SF2a figure and outline previous studies in lines 102 and 333 (Peacock et al., 2012). We also demonstrate that the full length AR does not regulate the seven genes and that full length AR is not associated with expression of the seven genes in human PC datasets (new figures SF3, SF7a, SF10, and SF11).

3. *In Figure 1B, what is the full 60 gene set that is regulated by AR-V7 knockdown? A table containing this in supplements would be very useful.*

We have now added a table containing the full 60 gene set in the new SF2b figure.

4. *As an aside, in Fig 1A, the grey module (#6, counting from top) and the red module (#9) are interesting in containing suppressor activities in Cancer/BPH/HGPIN.*

We thank the reviewer for pointing this out, and we plan to explore other WGCNA modules associated to PC in future research. For the present manuscript, we focused on the WGCNA module that was significantly enriched in AR-V7 regulated genes.

5. *In Figure 2A, the authors say "expression levels of the seven genes in human PC bone metastases"? Why only metastases? The authors should clarify what happens to "Cancer" and "CRPC" groups....the*

reviewer presumes this is the subdivision since nothing is clarified about the specifics of the analyses.

We apologize for our lack of clarity in this regard. The dataset consists only of CRPC bone metastases, and is one of the few dataset that contain information about the expression of AR-V7. We have clarified this point in lines 158, 160, and 719.

6. *Data for yeast selection is completely lacking - metrics for growth suppression or enhancement?? Colony pictures or the like?? Western blot for successful induction of AR-V7. Etc.*

We thank the reviewer for this helpful suggestion and addressed this in the new figure SF4 of the revised manuscript and in the Results and Methods sections.

We show here a Western Blot for successful induction (-thiamine) of AR-V7 in the yeast model.

7. *In Figure 3A and 3B significant details are missing. Concentrations ?? For how long? Endpoint? Surprisingly, panel (F) and (G) contain abundant details, which need to be moved to earlier panels.*

We have now divided Figure 3 into two separate figures and added more details to the figure legends. Figure 3 of the revised manuscript shows experiments in which there was individual stable depletion of the expression of the seven gene set; while Figure 4 explores the effects of pharmacologic inhibition of the gene set.

8. *In Figure 3C, the authors say "22Rv1 stably depleted of each of the seven genes were transfected with a dual plasmid luciferase reporter system which quantifies AR activity and basal transcription". Stably depleted? How do the cells survive? Is this a Doxycycline inducible? The system quantifies total AR*

activity. Then how is this specific to AR-V7? Are these genes generally controlled by AR?? Please explain.

We did not use a doxycycline inducible system, but we performed the reporter gene assays within 48 hours after selection of stably transduced cells, when the effects of gene depletion on cellular growth are minor. In addition, the dual reporter gene system used permits evaluation of basal transcription (a control plasmid in which the AREs are deleted from the MMTV promoter) and the intact MMTV promoter. We did not observe at 48 hours a decrease in general transcription or protein amounts following depletion of the seven genes as one would expect if the cells were not surviving. We now clarified in lines 203-205 that the assay was conducted in androgen-depleted conditions in 22Rv1, where AR ligand-independent activity is largely driven by AR variants including AR-V7 (Dehm et al., 2008; Guo et al., 2009). We also show in new SF10 that these genes are not controlled by full length AR.

Reviewer #2:

We thank the reviewer for the comment: “This is a well-written manuscript with significant data consistent with the conclusions drawn.”

...the prognostic gene module identified has very significant overlap with a previously identified prognostic tool, Cell Cycle Progression (CCP) score, which performs very well for prostate cancer, as well as some other cancers. Surprisingly, the authors did not even mention this in the manuscript.

We thank the reviewer for bringing the CCP gene signature to our attention and now include discussion of this signature in our manuscript (lines 286-293).

We compared the performance of our signature to the CCP, utilizing a simple cutoff threshold for two different prostate cancer datasets (TCGA and MSKCC). While our seven gene signature predicted disease-free survival and overall-survival for both datasets, the CCP signature was only able to predict disease-free survival in the TCGA dataset. A smaller signature such as the one we identified is easier to apply and obtain significant predictive and survival metrics. Our signature also provided the rationale for a

promising combination therapy to be explored for castration resistant prostate cancer.

Please also note that only three of the seven genes from our signature are found in the 31 gene CCP signature. Further, our seven gene signature was derived using a strong biological rationale and a systems-biology approach. In contrast, the 31 gene CCP signature was derived by choosing genes already well known to regulate cell cycle. The CCP gene signature is prognostic for prostate as well as other cancers. Whereas, the seven gene signature we developed is specific for survival metrics in prostate cancer alone, thus speaking to its unique disease characteristics (We have now expanded the number of other cancers and datasets as described below.)

1. *Line 96-100: The authors used a specific shRNA to knockdown AR-V7 in 22Rv1 cells. Experimental evidence, such as by qPCR and immunoblotting, should be provided to show that the effect is specific to the splicing variant, and not the full-length AR.*

We thank the reviewer for this suggestion and show that the shRNA for AR-V7 is specific for AR variants and does not affect the levels of full length AR (new figure SF2a, and Peacock et al., 2012).

2. *line 140-142: The gene expression dataset from the Homberg study is not a RNA-seq dataset.*

We have corrected this error.

3. *line 162-164: The authors state that expression of the 7-gene signature is prognostic specifically for prostate cancer. This is not warranted as only one cohort with a couple of types of cancer was used in the analysis. Comprehensive analysis of multiple cohorts of at least major types of cancer should be conducted to assess this possibility. (lung, colon, breast)*

We have now analyzed multiple cohorts and examined the four major cancer types. These data are shown in SF9. The seven gene signature was not prognostic for any of the other major types of cancer. We also examined two independent cohorts for breast cancer and found no association of the seven gene set with either cohort.

4. *There are very few samples (8) of CRPC included. This should be expanded.*

We agree with the reviewer on this point. However, there is no other dataset with CRPC samples on the same array platform to combine in the WGCNA analysis. Nevertheless, despite the low number of samples, the association of the expression levels of the genes within the green module to the CRPC stage is still significant. In addition, we utilized an independent CRPC dataset (Hornberg et al., 2011) to validate the association between AR-V7 and the seven genes.

5. *"Depleting CCNB2 did not significantly reduce ligand-independent AR activity as measured in either assay". In fact, CCNB2 depletion significantly increased AR activity. What could be the mechanism for this? Does it also affect full-length AR?*

We now show in SF14 that depletion of CCNB2 increased AR-V7 and full length AR mRNA levels. We address this point in the discussion, lines 313-321. Since the seven genes are interrelated and highly connected, it is possible that depletion of CCNB2 increases the expression of other members of the gene set in a compensatory way, which could drive the increase in AR-ligand independent transcriptional activity.

6. *In Fig. 3G, there appears to be at least additive effects for N9 vs N9+Dox in PC3 cells, while it is stated that there are no effects in this cell line. In addition, the combinatorial treatment should be tested in non-CRPC but AR-responsive cell lines, such as LNCaP and VCaP, to assess the specificity of the treatment for AR-V7 mediated growth.*

We thank the reviewer for the suggestions. The statistical analysis (performed against the vehicle treatment) shows that there are no significant differences between N9 and control group, or N9+Dox and control group in PC3 cells. We have now tested the single drug and combinatorial treatments in the AR-positive cell line LNCaP as the reviewer suggested (Figure 4b). We found that the combinatorial treatment had a far more modest antiproliferative effect on LNCaP compared to the CRPC cell lines.

7. *Again in Figure 3g, the combinatorial treatment have similar efficacy*

on an AR-V7 negative cell line (C4-2B) and AR-V7 positive cell line (22Rv1). This observation actually challenges the authors' statement that the 7-gene module is specific to AR-V7.

We thank the reviewer for raising this important point. While expression of the seven genes are selectively regulated by AR-V7 and not full-length AR (new figure SF10) two of the seven genes (*TOP2A*, *CCNB1*) encoded proteins that were previously shown to enhance full-length AR activity (Chen et al., 2006; Schaefer-Klein et al., 2015, Yu et al., 2014). Indeed, *TOP2A* and *CCNB1* are the targets of the two drugs we utilized. Thus, the effects of the drug combination on LNCaP and C4-2B are likely due to disrupting the enhancement of full length AR by *TOP2A* and *CCNB1*. We confirmed that depletion of *TOP2a* decreased full length AR levels (new figure SF14). Together these data support a more broad impact of the seven gene set and not solely to AR-V7 driven tumors.

8. The mechanisms of how the products of the seven genes identified can regulate AR-V7 activity are not explored. Is it on the expression of AR-V7, its stability, nuclear localization, etc.? Are the effects direct, or require additional factors?

We thank the reviewer for these suggestions. In response to the reviewer, we have begun these studies where we explored the effects of depleting one member of each category of genes (kinesins, cyclins, and mitotic checkpoints) of the gene set on AR-V7 levels. We found that individual depletion of *KIF20a*, *TOP2a*, and *BUB1b* decreased AR-V7 mRNA levels, while depletion of *CCNB2* increased AR-V7 mRNA levels (new figure SF14a). These findings are consistent with effects on AR-V7 signaling we have previously shown through reporter gene assays and expression of target genes, such as *FKBP5* and *UBE2C*. We also examined whether any of these members regulated full length AR. We found that individual depletion of *KIF20a* and *CCNB1* reduced full length AR mRNA levels, while depletion of *CCNB2* increased them (new figure SF14b). Depletion of *BUB1b* had no effects on full length AR levels. Since, as indicated by the reviewer, seven different genes may have a variety of effects on AR-V7 activity, we plan to pursue these experiments in greater depth in a separate manuscript.

9. *Since most prostate cancer patient tissue as well as the cell lines that express AR-V7 also express full length AR, it would be important to determine if the 7-gene model also affects the full length AR, and/or its heterodimerization with AR-V7.*

As discussed above, two of the proteins, CCNB1 and TOP2A enhance full length AR levels and/or activity. We acknowledge the possibility of effects of the seven encoded proteins on AR-V7:full length AR heterodimers. However, answering this question in a rigorous way would require extensive additional experimentation including bimolecular fluorescence complementation (BiFC) and bioluminescence resonance energy transfer (BRET) assays as conducted in the elegant studies of Xu et al., 2015. Regardless of the possible effects on AR full length:AR-V7 heterodimers, our data demonstrate that disruption of the seven genes affects CRPC proliferation and led to the identification of a synergistic combination of inhibitors.

Thank you for sending us your revised manuscript, related to your previous submission MSB-18-8202. We have now heard back from the reviewers who agreed to evaluate your manuscript. We have sent the manuscript back to reviewer #1 and to a new reviewer (#4), since reviewer #3 was not available this time. As you will see below, both reviewers mention that the study has improved as a result of the performed revisions and they are supportive of publication. However, reviewer #1 refers to the need to add some further discussion on related literature, which we would ask you to include in a minor revision.

Before we formally accept the manuscript for publication, we would also ask you to address a few remaining editorial issues listed below:

REFeree REPORTS

Reviewer #1:

The authors describe a gene signature associated with the activation of the prominent androgen-receptor splice variant AR-V7 in CRPC, and the prognostic value of this gene signature. In a novel approach, the authors show that screening for specific modulators of AR-V7 in the yeast *S. pombe* provides a support for this signature and extends the possibility of using this genetic system as a rapid, high-throughput, and low cost surrogate for screening genetic modulators of human proteins. The signature clearly shows specificity for prostate cancer. Preliminary analysis of therapeutic targeting of a combination of TOP2A and CCNB1 suggest that synergistic targeting of other proteins in this module, or other expanded modules, may be of value in CRPC with AR-V7 activation/expression.

General Impression: The manuscript in its revised form is significantly improved by the added data and experiments. The systems biology analyses is strong and comprehensive. These data lend additional support to the conclusion that this AR-V7 gene signature is significantly regulated by AR-V7 compared to AR-FL, although an interplay between AR-V7 and AR-FL cannot be ruled out. The novelty of this study lies in the use of the yeast genetic screen as a rapid, high-throughput, and low cost surrogate for use in human systems biology and the results are encouraging - even though its use may be restricted to specific biological circumstances. It would behoove the authors to highlight this aspect since a number of previous papers have presented gene expressions studies dissecting the contributions of AR-FL v AR-V7 in prostate cancer.

For example, the following papers mentioned below, which the authors would benefit from discussing in this manuscript, already explore the role of AR-V7 in prostate cancer. These are only examples, and others certainly exist. Indeed, one paper develops a signature for AR-V7 that subsets the genes highlighted in the authors current manuscript. Furthermore, two example studies show that the activity of AR-V7 may not be independent of AR-FL, and do heterodimerize. These considerations explain why in specific experiments the authors cannot fully dissect the AR-V7 contribution from AR-FL. Indeed, the two genes TOP2A and CCNB1 that were tested in drug studies reflect this interplay. These aspects should be discussed in the context of the authors' findings.

P. Watson, Y. Chen, M. Balbas, J. Wongvipat, N. Socci, A. Viale, et al. Constitutively active androgen receptor splice variants expressed in castration-resistant prostate cancer require full-length androgen receptor. *Proc Natl Acad Sci U S A*, 107 (2010), pp. 16759-16765.

D. Xu, Y. Zhan, Y. Qi, B. Cao, S. Bai, W. Xu, et al. Androgen receptor splice variants dimerize to transactivate target genes. *Cancer Res*, 75 (2015), pp. 3663-3671

Hu R, Lu C, Mostaghel EA, Yegnasubramanian S, Gurel M, Tannahill C, et al. Distinct transcriptional programs mediated by the ligand-dependent full-length androgen receptor and its splice variants in castration-resistant prostate cancer. *Cancer Res*. 2012 Jul 15;72(14):3457-62.

Reviewed for example in:

Lu J, Van der Steen T, Tindall DJ. Are androgen receptor variants a substitute for the full-length receptor? *Nat Rev Urol*. 2015 Mar;12(3):137-44.

In conclusion, this reviewer recommends publication of the paper, pending editorial decisions.

Reviewer #4:

The manuscript has addressed the major concerns of the previous critique, and will be of impact for the prostate cancer and nuclear receptor fields.

Response to reviewer #1:

Thank you for your valuable comments, and as it was suggested we included and discussed the recommended references. The following changes were made in the manuscript:

1. Page 3, paragraph 2, line 60: reference Watson et al., 2010 was included and the reference Chan et al., 2015 was moved from line 64 to line 60.
2. Page 4, paragraph 1, line 64: reference Imamura & Sadar, 2016 was added.
3. Page 4, paragraph 2, line 67: reference Hu et al., 2012 was included.
4. Page 4, paragraph 2, line 73-76: the sentence now reads: 'Identifying specific disease-relevant, AR-V7-driven genes is challenging, in part because of the extensive overlap with the full-length AR transcriptome' and the references Watson et al., 2010; Chan et al., 2012; Hu et al., 2012; reviewed in Lu et al., 2015 were included.
5. Page 12, paragraph 3, line 276- 277: the following text was added: 'consistent with Hu et al., (2012), who demonstrated AR-V7 regulation of several genes involved in mitosis'.
6. Page 13, paragraph 2, line 303: the reference Xu et al., 2015 was added.
7. Page 27, line 662-663: the reference was added, Imamura, K. & Sadar, M.D. (2016). Androgen receptor targeted therapies in castration-resistant prostate cancer: Bench to clinic. *Int J Urol.* Aug;23(8):654-65.
8. Page 28: line 697-698: the reference was added Lu, J., Van der Steen, T., Tindall, D.J. (2015). Are androgen receptor variants a substitute for the full-length receptor? *Nat Rev Urol.* Mar; 12(3): 137- 144.

Corresponding Author Name: Burnstein, Kerry L.
 Manuscript Number: MSB-18-8202R-Q